# Vascular Changes in the Macula of Patients after Previous Episodes of Vision Loss Due to Leber Hereditary Optic Neuropathy and Non-Arteritic Ischemic Optic Neuropathy

**DOI:** 10.3390/diagnostics13101726

**Published:** 2023-05-12

**Authors:** Katarzyna Nowomiejska, Patrycja Lesiuk, Agnieszka Brzozowska, Katarzyna Tońska, Robert Rejdak

**Affiliations:** 1Department of General and Pediatric Ophthalmology, Medical University of Lublin, 20-093 Lublin, Poland; 2Department of Informatics and Medical Biostatistics, Medical University of Lublin, 20-059 Lublin, Poland; 3Institute of Genetics and Biotechnology, Faculty of Biology, University of Warsaw, 02-096 Warsaw, Poland

**Keywords:** OCT-A, microvascular dysfunction, Leber hereditary optic neuropathy, non-arteritic ischemic optic neuropathy

## Abstract

Purpose: to assess the vasculature and thickness of the macula using OCT-A in patients who had experienced a previous episode of Leber hereditary optic neuropathy (LHON) or non-arteritic anterior ischemic optic neuropathy (NA-AION). Methods: twelve eyes with chronic LHON and ten eyes with chronic NA-AION and eight NA-AION fellow eyes were examined using OCT-A. The vessel density was measured in the superficial and deep plexus of the retina. Moreover, the full and inner thicknesses of the retina were assessed. Results: There were significant differences in all sectors between the groups in regard to the superficial vessel density and the inner and full thicknesses of the retina. The nasal sector of the macular superficial vessel density was affected more in LHON than in NA-AION; the same with the temporal sector of the retinal thickness. There were no significant differences between the groups in the deep vessel plexus. There were no significant differences between the vasculature of the inferior and superior hemifields of the macula in all groups and no correlation with the visual function. Conclusions: The superficial perfusion and structure of the macula assessed with OCT-A are affected both in chronic LHON and NA-AION, but more in LHON eyes, especially in the nasal and temporal sectors.

## 1. Introduction

Leber hereditary optic neuropathy (LHON) is a rare disease caused by mutations in both mitochondrial DNA-encoded and nuclear DNA-encoded mitochondrial gene mutations [1]. The majority of LHON cases are due to pathogenic variants in the mitochondrial genome (mtDNA) affecting the subunits of the mitochondrial respiratory chain complex I [2]. LHON is characterized by a painless and rapid unilateral decrease in the visual acuity and central scotoma usually in the second or third decade of life [3]. It is followed by the involvement of the second eye within a few weeks. It affects mainly males and manifests with circumpapillary telangiectatic capillaries [4]. The pathogenesis of LHON is still elusive and it is supposed that vascular alterations may play a role. Interestingly, optic disk blood flow might be influential on the risk of conversion, as previously hypothesized [5]. Peripapillary and optic nerve head microangiopathy is a hallmark feature of the acute stages of LHON. However, there are few reports describing the vasculature of the whole retina in patients with LHON [6,7,8,9,10].

Non-arteritic anterior ischemic optic neuropathy (NA-AION) is the leading cause of sudden vision loss in the middle-age group and the most common non-glaucomatous optic neuropathy [11]. The loss of vision due to NA-AION is irreversible. NA-ION occurs as a result of sudden ischemia of the retinal ganglion cell axons in the region of the laminar cribrosa due to impaired posterior ciliary artery circulation in the optic nerve head [12,13].

Both LHON and NA-AION share the mechanism of vascular abnormalities of the optic nerve in the acute phase, leading to profound central vision loss, although there is a different pathogenesis of these entities. In the chronic phase, it is difficult to distinguish between these neuropathies, as they both lead to the optic disc pallor. The chronic phase alterations of the macula in chronic LHON and NA-AION have not, so far, been investigated deeply.

Optical coherence tomography angiography (OCT-A) is a novel, non-invasive imaging technique for analyzing the retinal and choroidal microvasculature in vivo [14]. OCT-A has already been used in research studies and the clinical management of patients not only with ophthalmological diseases, but also in neuro-ophthalmology, neurology and psychiatry [15,16,17]. OCT-A is able to show a reduction in the blood flow of the optic nerve head in optic neuropathies of different origins [18,19]. Eyes with previous episodes of NA-AION and LHON have not been assessed so far in one study.

The aim of this study was to evaluate the vascularization of the macula using OCT-A in patients who had experienced a previous episode of NA-AION and LHON.

## 2. Materials and Methods

### 2.1. Study Type

This was a cross-sectional study. The study was conducted at the Chair and Department of General and Pediatric Ophthalmology of the Medical University of Lublin, Poland. Approval of the Ethics Committee of the Medical University of Lublin was obtained (nr KE-0245/203/10/2022). Written informed consent was obtained from all participants of this study.

### 2.2. Patient Characterisctics

Twelve eyes of twelve patients (11 males, 1 female; mean age 36 years) with molecularly confirmed LHON were included in the study. All patients presented the chronic phase of LHON according to Carelli [20] and Balducci [21], more than 12 months from vision loss onset. Moreover, 10 eyes with a history of NA-AION onset in the past were also included (2 patients with bilateral AION and 8 with monocular AION; 7 males, 3 females; mean age 60 years). NA-AION in their history was defined based on the following criteria: sudden and painless vision loss; optic disc edema, or a small amount of hemorrhage around the optic disc; defect of the visual field, with horizontal or vertical hemianopia; the absence of neurological diseases with central nervous system (CNS) imaging. Eight fellow eyes of patients with monocular NA-AION were taken into consideration as a separate group.

Inclusion criteria were as follows: a history of NA-AION or LHON in the past (more than 12 months) and pallor of the optic disc. Exclusion criteria were as follows: inflammatory, genetic, traumatic, and other causes of optic nerve disease, poor fixation during OCT-A examination, poor cooperation, history of intraocular surgery, laser, and medication in the past 6 months.

Visual acuity was examined with Snellen charts and then converted to the logarithm of the minimum angle of resolution (logMAR) scale. All patients underwent ophthalmological examination including slit-lamp examination and fundoscopy after mydriasis. Automated static perimetry (Humphrey, 30–2 SITA standard) was performed on the day of OCT-A analysis in all cases. Mean deviation (MD), pattern standard deviation (PSD) and visual field index parameters were taken into consideration.

The control group consisted of 20 eyes of 20 normal individuals of a similar age: 10 patient younger than 40 years (mean age 31 years) and 10 patients older than 40 years (mean age 51 years), with normal optic disc and no eye or systemic diseases. Controls were recruited from outpatients attending a routine ophthalmological examination and medical staff.

### 2.3. OCT-A Examination

OCT-A examination was performed by the same ophthalmologist using an Optovue (Fremont, CA, USA) device. The OCT-A scans were calculated using the integrated software algorithm; 70,000 A-scan/s. Identification and segmentation methods of areas were selected automatically. Inner thickness (internal limiting membrane [ILM]-inner plexiform layer [IPL]), full thickness (ILM-retinal pigment epithelium [RPE]) and vessel density in 4 quadrants (superficial and deep vascular complex) were measured with the same device on scans 6 × 6 mm for the macula. All sectors of the macula were analyzed (inferior, superior, nasal, temporal); moreover, comparison between superior and inferior hemifields of the macula was carried out in all the groups. Moreover, foveal avascular zone (FAZ) was assessed in mm^2^. All scans were reviewed independently by two investigators to ensure correct segmentation and sufficient imaging quality; the cut-off quality score of OCT-A scans was 7/10.

### 2.4. Statistical Analysis

The database and statistical computations were carried out with STATISTICA 13.0 computer software (StatSoft, Krakow, Poland). The values of the analyzed measurable parameters were presented as the mean, median, and standard deviation (SD) and as counts and percentage in the case of non-measurable parameters. Both eyes of each patient were tested, but only one eye was taken for analysis at random, as data from the right and left eyes of the same patient in binocular diseases are not independent but correlated. This is in agreement with suggestions made by Armstrong [22].

For the measurable features, the normal distribution of the analyzed parameters was assessed using the Shapiro–Wilk test. The Kruskal–Wallis test using multiple comparison analysis was used to compare multiple independent groups. The Mann–Whitney test was used for the comparison of two independent groups (LHON and NA-AION). A level of significance of *p* < 0.05 indicated the existence of statistically significant differences.

## 3. Results

### 3.1. Superficial Vessel Density

There were significant differences (*p* < 0.0001) in the superficial vessel density (%) between the groups (Table 1, Figure 1), apart from the fovea (*p* = 0.24). The lowest values of the whole and sectoral superficial vessel densities were obtained in LHON eyes. However, there were no significant differences (*p* > 0.05) between LHON and NA-AION eyes in macular superficial vessel density (Table 1), apart from the nasal sector (*p* = 0.01). Additional analysis showed that there were no significant differences (*p* < 0.05) between AION fellow eyes and other groups in the superficial vessel density.

### 3.2. Deep Vessel Density

There were no significant differences (*p* < 0.05) between the groups in regard to deep vessel density (Table 2, Figure 2). Moreover, there were no significant differences between the LHON and NA-AION groups (Table 2). Additional analysis showed significant differences (*p* = 0.05) between AION fellow eyes and the control group in the deep vessel density of the inferior and nasal quadrants, whole deep, inferior and superior hemifields and parafovea.

### 3.3. Inner Thickness of the Retina

There were significant differences (*p* < 0.0001) in all the values in the inner thickness of the retina between groups (Table 3). There was a significant difference (*p* = 0.03) between LHON and NA-AION eyes in regard to the temporal sector of the inner thickness of the macula (Figure 3). Additional analysis showed that there were no significant differences (*p* < 0.05) between AION fellow eyes and other groups in the inner thickness of the retina.

### 3.4. Full Thickness of the Retina and FAZ

There were also significant differences between the groups in regard to full retinal thickness (Table 4, Figure 4). Full retinal thickness in sectors and in the fovea was the lowest in LHON eyes. The differences were not significant for FAZ (*p* > 0.05) (Table 4). There were significant differences between LHON and NA-AION eyes in regard to inferior and temporal sectors of the full thickness of the macula (Table 4). Additional analysis showed no significant differences (*p* > 0.05) between AION fellow eyes and the other groups in the full thickness of the retina.

There were no significant correlations found between OCT-A parameters and visual acuity and visual field parameters.

### 3.5. Comparison of Superior and Inferior Hemifields of the Macula

Statistical analysis showed that, taking into account the absolute values of the differences between the superficial and deep vessel density of the superior and inferior hemifields, there were no significant differences found between the disease groups and the control groups; there were also no significant differences in regard to the full thickness and inner thickness of the retina (*p* > 0.05) (Table 5). Furthermore, there were no significant differences between the NA-AION and LHON groups in regard to the absolute values of the differences between superior and inferior hemifield (*p* < 0.05) obtained in the statistical analysis.

### 3.6. Visual Function Results

There were significant differences in regard to the visual acuity and the visual field parameters between groups. The median visual acuity in the LHON group was 1.1 logMAR (SD ± 0.63), 0.3 logMAR (SD ± 0.61) in NA-AION group and 0.0 logMAR (SD ± 0.06) in AION fellow eyes (*p* = 0.001). The median visual field index was in 65.5% in LHON eyes (SD ± 29.18%), 79% (SD ± 33.45%) in NA-AION eyes and 99% (SD ± 2.45%) in AION fellow eyes (*p* = 0.02). The median MD value was −11.62 dB (SD ± 8.97 dB) in LHON eyes, −8.76 dB (SD ± 9.92 dB) in NA-AION eyes and −1.54 dB (SD ± 1.25 dB) in AION fellow eyes (*p* = 0.06). The median PSD value was 8.68 dB (SD ± 3.18 dB) in LHON eyes, 8.95 dB (SD ± 4.25 dB) in NA--AION eyes and 2.3 dB (SD ± 0.95 dB) in AION fellow eyes (*p* = 0.01). Additional statistical analysis revealed statistically significant differences between NA-AION eyes and AION fellow eyes and between LHON eyes and AION fellow eyes in regard to all previously mentioned visual function parameters, but no differences between LHON and AION.

A qualitative assessment of the visual field results revealed visual field defects in the upper hemifield in four eyes, in the lower hemifield in five eyes and in the lower quadrant in three eyes.

### 3.7. Examples of Cases

The representative results of the four groups are shown in Figure 5.

## 4. Discussion

This is the first study to analyze macular vascular architecture in eyes after a previous episode of LHON and NA-AION. The major finding is that the vascular and structural alterations of the macula were present both in chronic LHON and NA-AION, although the vascular changes were more pronounced in LHON. The lowest values of the whole and sectoral superficial vessel density were obtained in LHON eyes. Full and inner thickness in the temporal sector of the macula were more decreased in chronic LHON than in chronic NA-AION.

Both NA-AION and LHON are optic neuropathies that have a similar clinical course in the natural history of the disease, there is rapid vision loss and a subsequent chronic phase with optic disc pallor and stable visual acuity. However, the pathogenesis remains different; for LHON, it is mitochondrial disfunction of retinal ganglion cells which may lead to apoptotic cell death and axonal degeneration, resulting in optic atrophy [4]. In NA-AION, there is ischemia of the retinal ganglion cells due to impaired blood flow in the posterior ciliary arteries [12]. For both diseases, there is still a lack of an effective treatment that can dramatically restore the vision.

Optic disc perfusion had already been assessed in patients with acute NA-AION [21]. In acutely affected eyes, OCT-A revealed a reduction in the signal from the major retinal vessels and the dilation of the patient’s superficial capillaries in the peripapillary area. However, non-acutely affected eyes showed attenuation of the patient’s capillaries [23].

In patients with LHON, OCT-A revealed the loss of peripapillary vessels and increased vessel tortuosity of the remaining ones, which leads to a reduction in the blood flow of the optic nerve head and eventually to loss of nerve fibers and disc atrophy [8]. Kousal [7] and colleagues examined 12 eyes of 6 patients who had experienced a previous episode of vision loss due to LHON; however, only peripapillary vessels were assessed. Balducci [20] found that the RPC vessel density in the peripapillary area was reduced in the chronic stage of LHON, and Borrelli [9] showed that the parafoveal vessel density in the macular area was lower in 29 eyes of LHON patients than in the controls. Fellow AION eyes were significantly different from the control group in regard to the majority of the parameters only in the deep macular vascular plexus. It may be anticipated that this plexus might be more vulnerable to ischemia than the superficial one in anatomically predisposed eyes.

In a recent large study, Castillo and co-workers [6] assessed 119 eyes of 60 patients with different stages of LHON. There was a general reduction in vessel density as the disease progressed, but with different patterns in the macular and the circumpapillary regions. Yu and colleagues [24] assessed the parafoveal and peripapillary areas in 14 LHON patients. In the chronic LHON patients, the superficial capillary plexus and inner retina thicknesses were significantly lower in all sectors than in the controls; the radial peripapillary capillary network vessel density and thickness were significantly lower in all sectors than in the controls and lower in the temporal, superior temporal, inferior temporal, and nasal sectors than in the subacute-stage LHON patients.

Laser speckle flowgraphy was also used to assess the retinal blood flow in LHON patients [25]. The major finding was that the optic disks of patients with LHON within 3 months from clinical onset showed high blood flow. In chronic LHON, there was delayed retinal nerve fiber layer (RNFL) thinning after combined ganglion cell layer + inner plexiform layer (GCIPL) thinning and blood flow loss.

It has already been shown that there is superficial peripapillary capillary dilation in eyes with acute AAION [26]. Rougier and colleagues [27] additionally studied the RNFL segmentation visualizing the peripapillary network as the edema-related darkening was reduced. They concluded that the increased capillary flux index, defined as the total weighted area of perfused vasculature per unit area, suggested an autoregulatory phenomenon to compensate for the ischemic process at the ciliary vasculature. Macular hypoperfusion and a significant thinning of the inner retinal layers (mainly the ganglion cell and inner plexiform layers) have also been described in NA-AION [28].

Fard and colleagues [29] compared macular and parafoveal vasculature in patients with optic disc swelling due to NA-AION. The macular and parafoveal superficial and deep vasculature density values were significantly lower in NAION eyes than in control eyes before detectable macular ganglion cell atrophy. None of the vessel density values were statistically different between papilledema eyes and control eyes.

In a recent study by Petrovic Pajic and colleagues [30], the retinal ganglion cell complex thickness in LHON eyes appeared to be relatively preserved in the central ETDRS circle compared to nonLHON optic neuropathies in the chronic phase. The nonLHON group in this study consisted of participants in whom no pathogenic variants were found in either mtDNA or nDNA, but their phenotype included bilateral optic neuropathy without any other detectable cause. The authors anticipate that in LHON the impact of mitochondrial dysfunction on retinal ganglion cell apoptosis and axonal degeneration may have a different gradient from the center to the periphery than in other optic neuropathies.

The present study adds additional knowledge to the pathogenesis and natural history of LHON and NA-AION. It has already been shown that ganglion cell layer and inner plexiform layer are acutely unaffected in NA-AION after one month from the episode of vision loss [31]. The thinning of the ganglion cell layer develops within 1 to 2 months of onset, which is prior to RNFL swelling resolution. Knowing that approximately 50% of the retinal ganglion cells are located in the human macula region and the central 10° of the visual field is affected in NA-AION, exploring the macula region seems to be useful for detecting structural loss. Interestingly, in another optic neuropathy, optic neuritis, there is a thinning of the retina ganglion cell layer plus the inner plexiform layer in the one-to-two month period after an episode of optic neuritis [32].

Both LHON and NA-AION lead to optic disc atrophy in the chronic stage. Our study showed that, in the chronic phase of both LHON and NA-AION, there were changes in the deep vascular plexus and the retinal thickness, different in both groups. Thinning of the retinal thickness was more pronounced in chronic LHON eyes (in the temporal sector) and a reduction in the superficial vascular plexus was more pronounced in chronic NA-AION eyes. These changes are probably due to the reduction in the overall metabolic rate of the retina in the chronic phase of these optic neuropathies. Thinning of the full and the inner thickness of the macula represents permanent loss of inner retinal neurons. Although the clinical picture of both neuropathies is similar after some time, the changes in the macula obtained with OCT-A may be helpful to differentiate between these two entities in the chronic phase. Changes in the macula detected using OCT-A- may be useful in future clinical trials regarding optic neuropathies as the possible objective endpoint.

Interestingly, there were no significant differences between upper and lower hemifields in either chronic NA-AION or LHON found in our study, despite the differences in visual field impairment in the upper and lower hemifields. It is already known that in the acute phase of NA-AION, there is a vascular density decrease with visual field defects and RNFL loss matching [33]. However, it seems that in the chronic phase of both NA-AION and LHON, there is a diffuse loss of vessels in the macula, being the retinal area with the highest energy demand. The defect of retinal vascularization is more pronounced in the deep plexus of NA-AION eyes, originally vascular disease. Special attention should be paid also to the macula of LHON patients. It has been speculated that cone photoreceptors, the retinal cells with the greatest oxygen consumption and characterized by a low mitochondrial reserve capacity, may be more vulnerable to the mitochondrial dysfunction caused by mtDNA mutations [34].

Fard and colleagues [35] compared the results of OCT-A in the chronic stage of two neuropathies: demyelinating optic neuritis and AION. They concluded that OCT-A data have a limited role in differentiating these disorders in the chronic phase.

The nature of LHON has already been assessed using choroidal thickness in the study by Borelli and co-workers [36]. They found that macular and peripapillary choroidal thicknesses were significantly increased in the acute LHON stage. In contrast, macular choroidal thickness was significantly reduced in the chronic stage.

Retinal blood flow has already been recognized as a potentially valuable biomarker for neurodegenerative optic neuropathies [20]. It has already been shown that retinal microvascular changes in LHON appear earlier in the macula than in the circumpapillary region [6]. There is a possible reduction in the metabolic rate of the retina in LHON, accompanied by a corresponding reduction in the retinal flow. Another hypothesis is that the vascular blood flow is primarily reduced in LHON. It suggests that vascular factors may play an important role in the mechanism of LHON, even in a chronic phase of the disease. This may have further implications for the future therapeutic options and end-points in clinical trials.

The shortcoming of our study is that the groups were relatively small, but when rare diseases such as LHON are considered, only multicenter studies give the possibility of larger groups of patients. As this was a cross-sectional study, longitudinal studies are further required to monitor the changes in the vasculature of optic neuropathies over time.

## 5. Conclusions

Our study has shown that the vasculature of the macula is decreased both in LHON and in NA-AION eyes in the chronic phase of the diseases, with some differences, especially in regard to the superficial vascular plexus. OCT-A may be an additional valuable diagnostic tool for detecting microvascular changes in the retina and to provide an objective outcome measure for assessing response to future therapies, both in LHON and NA-AION.

## Figures and Tables

**Figure 1 diagnostics-13-01726-f001:**
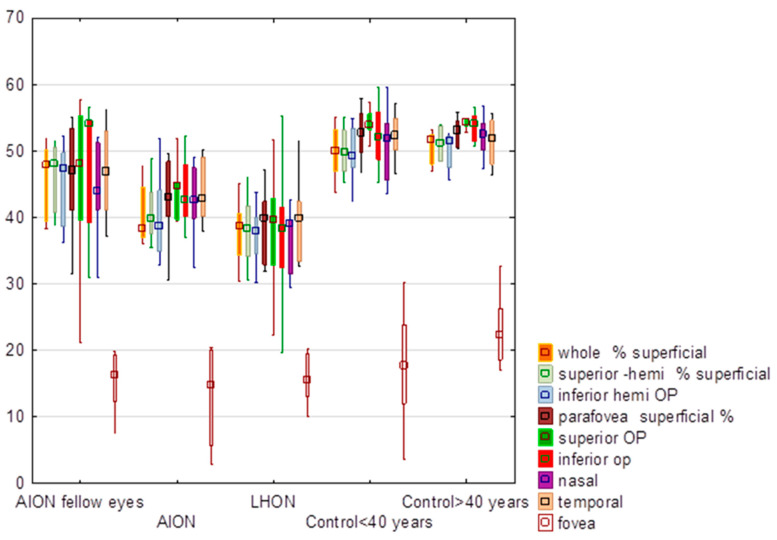
Comparison of values of the superficial vessel density (%) in eyes with anterior ischemic optic neuropathy (AION), AION fellow eyes, Leber hereditary optic neuropathy (LHON), control group of subjects younger than 40 years and older than 40 years. In the graph values are expressed in medians; boxes mean 25–75%; whiskers mean the range of non-outliers.

**Figure 2 diagnostics-13-01726-f002:**
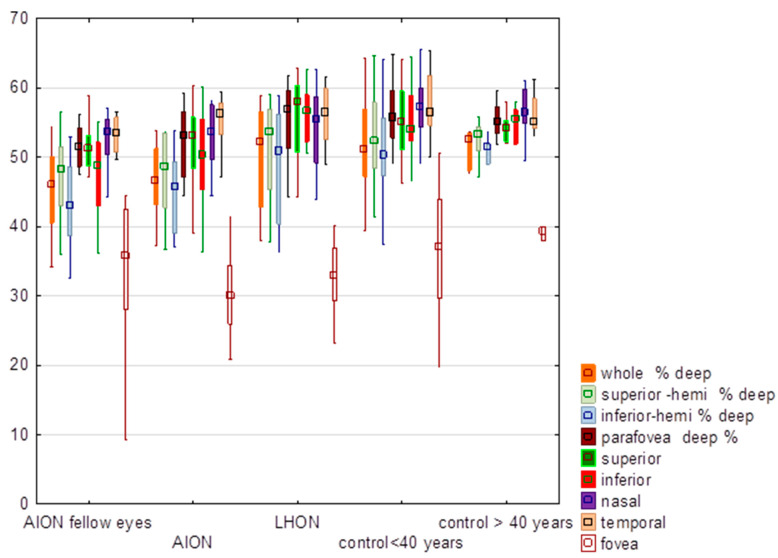
Comparison of values of the deep vessel density (%) in eyes with anterior ischemic optic neuropathy (AION), AION fellow eyes, Leber hereditary optic neuropathy (LHON), control group of subjects younger than 40 years and older than 40 years. In the graph values are expressed in medians; boxes mean 25–75%; whiskers mean the range of non-outliers.

**Figure 3 diagnostics-13-01726-f003:**
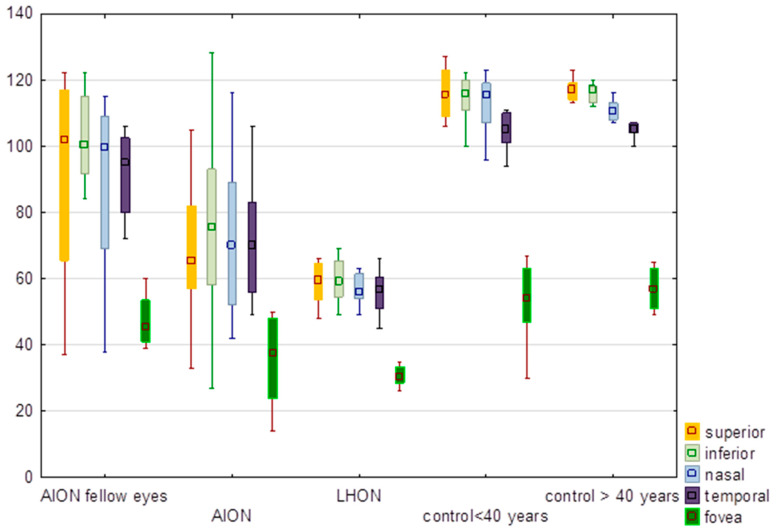
Comparison of values of the inner thickness of the macula (µm) in eyes with anterior ischemic optic neuropathy (AION), AION fellow eyes, Leber hereditary optic neuropathy (LHON), control group of subjects younger than 40 years and older than 40 years. In the graph values are expressed in medians; boxes mean 25–75%; whiskers mean the range of non-outliers.

**Figure 4 diagnostics-13-01726-f004:**
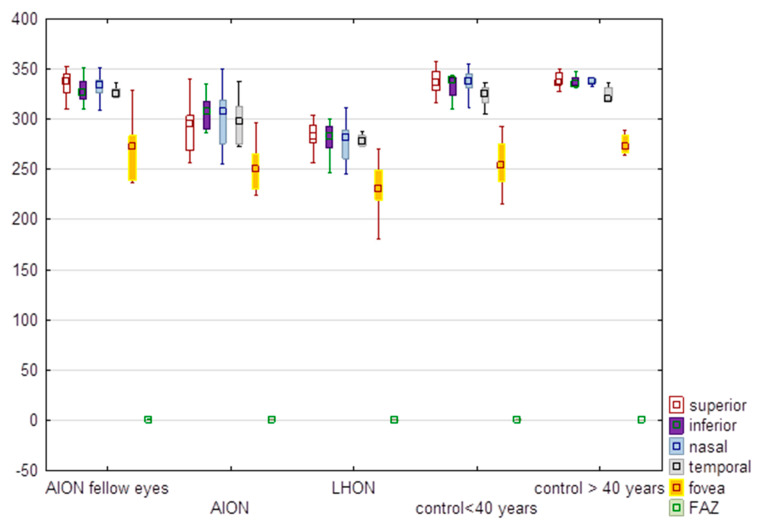
Comparison of values of the full thickness of the macula (µm) in eyes with anterior ischemic optic neuropathy (AION), AION fellow eyes, Leber hereditary optic neuropathy (LHON), control group of subjects younger than 40 years and older than 40 years. FAZ-foveal avascular zone. In the graph values are expressed in medians; boxes mean 25–75%; whiskers mean the range of non-outliers.

**Figure 5 diagnostics-13-01726-f005:**
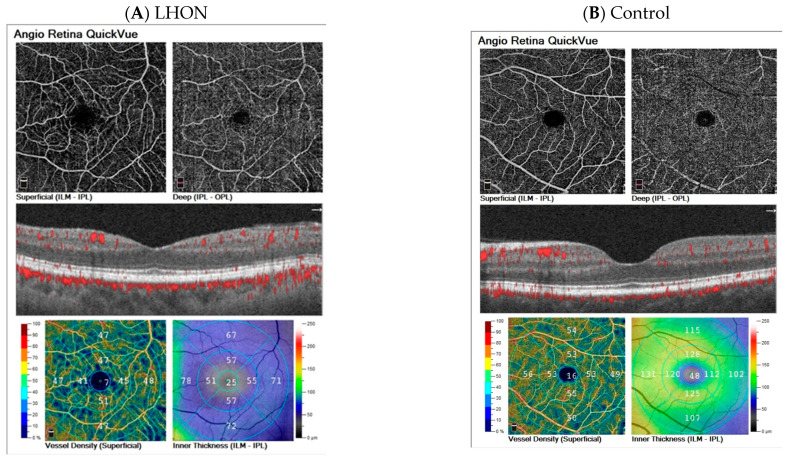
Representative OCT-A results of Leber hereditary optic neuropathy (LHON)—(**A**), control group—(**B**), anterior ischemic optic neuropathy (AION)—(**C**) and fellow eye of AION—(**D**).

**Table 1 diagnostics-13-01726-t001:** Vessel density (%) in the macular superficial vascular complexes across groups and results of statistical analysis. Asterisk (*) means statistical significance (*p* < 0.05).

Parameter	AION			AION Fellow Eye			LHON			Control < 40 Years			Control > 40 Years			Statistical Analysis		
	Mean	Median	Standard Deviation	Mean	Median	Standard Deviation	Mean	Median	Standard Deviation	Mean	Median	Standard Deviation	Mean	Median	Standard Deviation	Kruskal–Wallis Test	Mann–Whitney Test for LHON and NA-AION Z	*p*
Whole superficial	40.35	38.30	4.48	45.60	47.75	5.63	36.41	38.65	8.39	50.05	50.05	3.48	50.65	51.65	2.55	H = 29.63, *p* < 0.0001 *	−0.79	0.43
Superior hemi superficial	40.92	39.85	4.20	46.13	48.10	5.22	36.83	38.40	8.67	50.18	49.75	3.49	51.15	51.15	2.52	H = 29.96, *p* < 0.0001 *	−0.92	0.36
Inferior hemi	39.74	38.65	5.98	45.00	47.35	6.23	35.91	37.85	8.16	49.90	49.15	3.67	50.15	51.45	2.92	H = 27.75, *p* < 0.0001 *	−0.69	0.49
Parafovea superficial	42.67	42.90	5.82	46.23	47.10	8.24	37.19	39.80	8.71	52.42	52.75	3.69	52.92	53.10	2.14	H = 28.46, *p* < 0.0001 *	−1.62	0.11
Superior	42.13	44.75	9.55	45.55	47.95	12.26	38.35	39.65	7.59	53.83	53.95	3.01	54.37	54.35	1.17	H = 27.36, *p* < 0.0001 *	−1.68	0.09
Inferior	42.00	42.60	9.43	47.86	54.00	10.01	37.98	38.35	8.97	52.36	52.20	4.52	53.73	54.15	2.26	H = 20.98, *p* = 0.0001 *	−1.52	0.13
Nasal	42.63	42.65	4.90	44.48	44.00	7.13	33.98	39.10	11.37	51.15	51.90	5.33	52.22	52.50	3.39	H = 31.68, *p* < 0.0001 *	−2.80	0.01
Temporal	43.89	42.80	4.32	46.91	46.95	6.97	38.28	39.95	9.40	52.41	52.30	3.11	51.43	51.95	3.64	H = 25.11, *p* < 0.0001 *	−1.58	0.11
Fovea	11.75	13.15	6.97	16.96	16.40	7.27	14.75	14.15	3.71	20.37	21.90	5.49	21.38	22.10	2.80	H = 15.78 *p* = 0.003 *	0.88	0.38

**Table 2 diagnostics-13-01726-t002:** Values of the vessel density (%) in the macular deep vascular complexes across groups and results of statistical analysis.

Parameter	AION			AION Fellow Eye			LHON			Control < 40 Years			Control > 40 Years			Statistical Analysis		
	Mean	Median	Standard Deviation	Mean	Median	Standard Deviation	Mean	Median	Standard Deviation	Mean	Median	Standard Deviation	Mean	Median	Standard Deviation	Kruskal–Wallis Test	Mann–Whitney Test for LHON and NA-AION Z	*p*
Whole % deep	46.15	46.50	5.87	45.24	46.05	6.84	47.63	52.10	13.54	51.57	51.10	6.36	51.33	52.60	2.74	H = 8.06 *p* = 0.15	1.45	0.15
Superior-hemi % deep	47.48	48.65	6.37	47.23	48.20	6.57	48.94	53.65	13.11	52.46	52.30	6.40	52.48	53.30	3.05	H = 7.37 *p* = 0.19	1.42	0.16
Inferior-hemi % deep	44.72	45.65	5.84	43.23	42.95	7.01	46.14	50.75	14.53	50.67	50.35	6.53	50.25	51.30	3.28	H = 7.99 *p* = 0.16	1.42	0.16
Parafovea deep %	52.46	53.05	5.07	50.61	51.40	5.09	51.93	56.70	14.39	56.10	55.70	4.47	55.43	55.20	2.77	H = 8.29 *p* = 0.14	0.96	0.34
Superior	52.17	53.15	5.90	50.66	51.15	5.33	53.53	57.90	11.62	55.39	55.05	5.16	54.32	54.20	2.21	H = 8.09 *p* = 0.15	1.42	0.16
Inferior	50.05	50.35	7.64	47.38	48.75	6.39	51.72	56.60	14.64	54.73	54.05	5.40	54.90	55.45	2.67	H = 9.71 *p* = 0.08	1.38	0.17
Nasal	53.05	53.55	4.47	52.50	53.65	4.20	50.11	55.40	16.48	57.05	57.25	4.02	56.33	56.40	4.09	H = 7.44. *p* = 0.19	0.56	0.58
Temporal	54.57	56.20	4.88	51.95	53.40	5.51	52.48	56.50	15.09	57.29	56.40	4.07	56.17	55.15	3.10	H = 7.67 *p* = 0.17	0.63	0.53
Fovea	30.20	29.90	6.26	33.23	35.70	11.73	31.28	32.90	8.26	36.44	37.00	9.15	39.33	39.30	6.18	H = 7.46 *p* = 0.19	0.82	0.41

**Table 3 diagnostics-13-01726-t003:** Values of the inner thickness (µm) of the macula across groups and results of statistical analysis. Asterisk (*) means statistical significance (*p* < 0.05).

Parameter	AION			AION Fellow Eye			LHON			Control < 40 Years			Control > 40 Years			Statistical Analysis		
	Mean	Median	Standard Deviation	Mean	Median	Standard Deviation	Mean	Median	Standard Deviation	Mean	Median	Standard Deviation	Mean	Median	Standard Deviation	Kruskal–Wallis Test	Mann–Whitney Test for LHON and NA-AION Z	*p*
Superior	292.60	295.50	24.71	335.00	337.00	14.15	283.08	282.50	14.02	338.00	336.00	13.20	338.67	336.50	7.99	H = 33.29. *p* < 0.0001 *	−1.12	0.26
Inferior	309.60	307.00	24.02	328.50	326.00	13.24	280.25	283.00	14.64	333.79	338.50	11.17	337.33	334.50	6.38	H = 30.97. *p* < 0.0001 *	−2.90	0.004 *
Nasal	302.70	307.50	28.86	332.38	333.50	12.84	277.67	281.50	20.79	336.43	338.00	13.21	339.50	337.50	6.66	H = 30.59. *p* < 0.0001 *	−1.95	0.05
Temporal	298.60	297.00	23.43	323.13	324.50	12.52	276.25	278.00	12.61	323.79	325.50	10.34	324.50	320.50	7.97	H = 27.76. *p* < 0.0001 *	−2.01	0.04 *
Fovea	253.20	250.00	22.69	269.75	273.00	31.44	229.83	230.00	27.68	253.14	254.00	22.78	274.83	272.50	9.75	H = 14.48. *p* = 0.006 *	−1.85	0.06

**Table 4 diagnostics-13-01726-t004:** Values of the full thickness (µm) of the macula across groups and results of statistical analysis. Asterisk (*) means statistical significance (*p* < 0.05).

Parameter	AION			AION Fellow Eye			LHON			Control < 40 Years			Control > 40 Years			Statistical Analysis		
	Mean	Median	Standard Deviation	Mean	Median	Standard Deviation	Mean	Median	Standard Deviation	Mean	Median	Standard Deviation	Mean	Median	Standard Deviation	Kruskall–Wallis Test	Mann–Whitney Test for LHON and NA-AION Z	*p*
Superior	292.60	295.50	24.71	335.00	337.00	14.15	283.08	282.50	14.02	338.00	336.00	13.20	338.67	336.50	7.99	H = 33.29. *p* < 0.0001 *	−1.12	0.26
Inferior	309.60	307.00	24.02	328.50	326.00	13.24	280.25	283.00	14.64	333.79	338.50	11.17	337.33	334.50	6.38	H = 30.97. *p* < 0.0001 *	−2.90	0.004 *
Nasal	302.70	307.50	28.86	332.38	333.50	12.84	277.67	281.50	20.79	336.43	338.00	13.21	339.50	337.50	6.66	H = 30.59. *p* < 0.0001 *	−1.95	0.05
Temporal	298.60	297.00	23.43	323.13	324.50	12.52	276.25	278.00	12.61	323.79	325.50	10.34	324.50	320.50	7.97	H = 27.76. *p* < 0.0001 *	−2.01	0.04 *
Fovea	253.20	250.00	22.69	269.75	273.00	31.44	229.83	230.00	27.68	253.14	254.00	22.78	274.83	272.50	9.75	H = 14.48. *p* = 0.006 *	−1.85	0.06
FAZ	0.30	0.30	0.08	0.28	0.29	0.08	0.28	0.28	0.07	0.29	0.30	0.11	0.23	0.22	0.09	H = 3.03. *p* = 0.55	−0.69	0.49

**Table 5 diagnostics-13-01726-t005:** Comparison of the absolute values of the differences between superior and inferior hemifields of the macula in the superficial and deep vessel density (%), as well as inner thickness and full thickness of the macula (µm) across groups. Asterisk (*) means statistical significance (*p* < 0.05).

Parameter	NA-AION			AION Fellow Eyes			LHON			Control Group		
	Mean	Median	Standard Deviation	Mean	Median	Standard Deviation	Mean	Median	Standard Deviation	Mean	Median	Standard Deviation
Vessel density Superficial	Z = 1.10; *p* = 0.92			Z = 0.85; *p* = 0.40			Z = 0.31; *p* = 0.75			Z = 2.35; *p* = 0.02 *		
Superior	42.13	44.75	9.55	45.55	42.13	44.75	9.55	45.55	42.13	44.75	9.55	45.55
Inferior	42.00	42.60	9.43	47.86	42.00	42.60	9.43	47.86	42.00	42.60	9.43	47.86
Absolute value of the difference between hemifields	3.35	2.70	2.43	5.56	3.35	2.70	2.43	5.56	3.35	2.70	2.43	5.56
Statistical analysis	H = 5.42. *p* = 0.14											
Vessel density deep	Z = 1.48; *p* = 0.14			Z = 2.10; *p* = 0.02 *			Z = 1.84; *p* = 0.07			Z = 0.50; *p* = 0.61		
Superior	52.17	53.15	5.90	50.66	52.17	53.15	5.90	50.66	52.17	53.15	5.90	50.66
Inferior	50.05	50.35	7.64	47.38	50.05	50.35	7.64	47.38	50.05	50.35	7.64	47.38
Absolute value of the difference between hemifields	4.70	2.90	4.62	3.44	4.70	2.90	4.62	3.44	4.70	2.90	4.62	3.44
Statistical analysis	H = 8.96. *p* = 0.03 *; no differences between groups											
Inner thickness	Z = 0.95; *p* = 0.34			Z = 0.42; *p* = 0.67			Z = 0.67; *p* = 0.50			Z = 1.41; *p* = 0.16		
Superior	71.90	65.50	26.09	91.00	71.90	65.50	26.09	91.00	71.90	65.50	26.09	91.00
Inferior	76.20	75.50	27.58	96.13	76.20	75.50	27.58	96.13	76.20	75.50	27.58	96.13
Absolute value of the difference between hemifields	11.70	6.50	11.51	9.88	11.70	6.50	11.51	9.88	11.70	6.50	11.51	9.88
Statistical analysis		H = 10.13. *p* = 0.02 *; no differences between groups										
Full thickness	Z = 1.17; *p* = 0.24			Z = 1.86; *p* = 0.06			Z = 1.83; *p* = 0.07			Z = 2.24; *p* = 0.03 *		
Superior	292.60	295.50	24.71	335.00	292.60	295.50	24.71	335.00	292.60	295.50	24.71	335.00
Inferior	309.60	307.00	24.02	328.50	309.60	307.00	24.02	328.50	309.60	307.00	24.02	328.50
Absolute value of the difference between hemifields	20.60	6.50	32.70	7.50	20.60	6.50	32.70	7.50	20.60	6.50	32.70	7.50
Statistical analysis	H = 3.68. *p* = 0.30											

## Data Availability

Not applicable.

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
