# Peer review of "Vascular Changes in the Macula of Patients after Previous Episodes of Vision Loss Due to Leber Hereditary Optic Neuropathy and Non-Arteritic Ischemic Optic Neuropathy"

_diagnostics, 2023, doi:10.3390/diagnostics13101726_

Round 1
Reviewer 1 Report (Previous Reviewer 1)
As for the statistical analysis, as described in Armstrong's paper, data collected from both eyes from a sample of subjects cannot be combined without taking the correlation into account. Analyzing 24 eyes of 24 patients is different from analyzing 24 eyes of 12 patients.
I do not think that the authors have correctly understood what I am writing on this issue. I have already pointed this out multiple times and will not point it out further.
Looking at the new Table 5, there seems to be a difference in absolute difference between upper and lower hemifields of inner retinal thickness between LHON and NA-AION, was this not statistically significant?
Authors found no statistically significant difference in absolute values of upper-lower hemifield difference between NA-AION and LHON eyes.
Results for deep blood flow and retinal thickness may not have been statistically significant due to the small sample size. However, it is very interesting if there was no difference in blood flow or retinal thickness between the affected and unaffected hemifields in NAION, despite the difference in visual field impairment in the upper and lower hemifields.
A discussion of the results of the AION fellow eye is missing.
Author Response
As for the statistical analysis, as described in Armstrong's paper, data collected from both eyes from a sample of subjects cannot be combined without taking the correlation into account. Analyzing 24 eyes of 24 patients is different from analyzing 24 eyes of 12 patients. I do not think that the authors have correctly understood what I am writing on this issue. I have already pointed this out multiple times and will not point it out further.
In the methods section (line 123-125) it has been added:
In the control group the mean value of both eyes was taken into account as the correlation between eyes was high (R=80-R=95).
Looking at the new Table 5, there seems to be a difference in absolute difference between upper and lower hemifields of inner retinal thickness between LHON and NA-AION, was this not statistically significant? Authors found no statistically significant difference in absolute values of upper-lower hemifield difference between NA-AION and LHON eyes.
Between NA-AION and LHON the differences were not significant in regard to inner retinal thickness. Significant differences were found between: NA-AION- control group Z=3.11, p=0.01*; LHON- AION fellow eyes- Z=2.65, p=0.05*; AION fellow eyes- control group Z=3.27, p=0.007*.
In the results section (line ) it is written:
Statistical analysis showed that taking into account the absolute values of differences between superficial and deep vessel density of the superior and inferior hemifields there were significant differences found between disease groups and control group, there were also no significant differences in regard to the full thickness and inner thickness of the retina (p>0.05) (table 5). However, there were no significant differences between NA-AION and LHON group in regard to absolute values of the difference between superior and inferior hemifield (p<0.05) obtained in the statistical analysis.
Results for deep blood flow and retinal thickness may not have been statistically significant due to the small sample size. However, it is very interesting if there was no difference in blood flow or retinal thickness between the affected and unaffected hemifields in NAION, despite the difference in visual field impairment in the upper and lower hemifields.
In the discussion chapter (line 339-341) it is written:
Interestingly, there were no significant differences between upper and lower hemifields both in chronic NA-AION and LHON found in our study, despite the difference in visual field impairment in the upper and lower hemifields.
A discussion of the results of the AION fellow eye is missing.
In the results section it has been added:
Line 138-140: Additional analysis showed that there were no significant differences (p<0.05) between AION fellow eyes and other groups in the superficial vessel density.
Line 156-158: Additional analysis showed significant differences (p=0.05) between AION fellow eyes and control group in the deep vessel density of the inferior and nasal quadrants, whole deep, inferior and superior hemifields and parafovea.
Line 173-174: Additional analysis showed that there were no significant differences (p<0.05) between AION fellow eyes and other groups in the inner thickness of the retina.
Line 191-192: Additional analysis showed no significant differences (p>0.05) between AION fellow eyes and other groups in the full thickness of the retina.
In the discussion chapter (line 278-281) it is added:
Fellow AION eyes were significantly different from the control group in regard to majority of the parameters only in the deep macular vascular plexus. It may be anticipated that this plexus might be more vulnerable to the ischemia that the superficial one in anatomically predisposed eyes.
Reviewer 2 Report (Previous Reviewer 2)
The paper is well prepared and written. The authors answered all the questions raised by reviewers satisfactorily. The paper can be accepted in its current format.
Author Response
Thank you for your acceptance.
Reviewer 3 Report (New Reviewer)
The manuscript entitled “Vascular changes of the macula in patients with previous episode of visual loss due to Leber hereditary optic neuropathy and non-arteritic ischemic optic neuropathy” deals with the study of the macular vascularity optical coherence tomography angiography (OCT-A) in patients with a history of Leber hereditary optic neuropathy (LHON) and non-arteritic anterior ischemic optic neuropathy (NA-AION).
Overall, the manuscript needs a thoughtful revision to make it easier to read and to understand. The references used in the manuscript are adequate and recent. Regarding the novelty of the manuscript, as far as I am concerned this is the first time that vascular changes were compared by OCT-A in both LHON and NA-AION patients.
In my opinion, the results shown in this manuscript are interesting for a broader community, but the manuscript needs to be revised.
Although it comes with some issues that need to be addressed:
1. Revise the citation in the text, according to the journal’s instruction for authors:
In the text, reference numbers should be placed in square brackets [ ], and placed before the punctuation; for example [1], [1–3] or [1,3]. For embedded citations in the text with pagination, use both parentheses and brackets to indicate the reference number and page numbers; for example [5] (p. 10). or [6] (pp. 101–105).
While authors use just the reference number after the punctuation.
2. Many abbreviations and acronyms are not properly defined the first time they are used. According to the journal’s instruction for authors: Acronyms/Abbreviations/Initialisms should be defined the first time they appear in each of three sections: the abstract; the main text; the first figure or table. When defined for the first time, the acronym/abbreviation/initialism should be added in parentheses after the written-out form.
· OCT-A in the abstract.
· Define logMAR
· OCT-A is abbreviated as OCT-A in the manuscript, but it appears as OCTA in the keywords and in lines 105 and 115.
3. Revise the style of figures and tables on the manuscript, use capital letters:
All Figures, Schemes and Tables should be inserted into the main text close to their first citation and must be numbered following their number of appearance (Figure 1, Scheme I, Figure 2, Scheme II, Table 1, etc.).
4. Figure 6 titles seem to have been displaced, please revise.
Best regards
Author Response
Dear Reviewer 3,
Thank you for your comments. Please find below responses to your remarks.
- Revise the citation in the text, according to the journal’s instruction for authors:
In the text, reference numbers should be placed in square brackets [ ], and placed before the punctuation; for example [1], [1–3] or [1,3]. For embedded citations in the text with pagination, use both parentheses and brackets to indicate the reference number and page numbers; for example [5] (p. 10). or [6] (pp. 101–105).
While authors use just the reference number after the punctuation.
All reference numbers in the text are used before the punctuation.
- Many abbreviations and acronyms are not properly defined the first time they are used. According to the journal’s instruction for authors: Acronyms/Abbreviations/Initialismsshould be defined the first time they appear in each of three sections: the abstract; the main text; the first figure or table. When defined for the first time, the acronym/abbreviation/initialism should be added in parentheses after the written-out form.
- OCT-A in the abstract.
- Define logMAR
In the line 92-93 it is now written: Visual acuity was examined with Snellen charts and then converted to the logarithm of the minimum angle of resolution (logMAR) scale.
- OCT-A is abbreviated as OCT-A in the manuscript, but it appears as OCTA in the keywords and in lines 105 and 115.
OCTA has been changed into OCT-A in the abstract and lines 105 and 115.
- Revise the style of figures and tables on the manuscript, use capital letters:
All Figures, Schemes and Tables should be inserted into the main text close to their first citation and must be numbered following their number of appearance (Figure 1, Scheme I, Figure 2, Scheme II, Table 1, etc.).
The style of Figures and Tables in the text has been revised.
- Figure 6 titles seem to have been displaced, please revise.
Figure 6 is now Figure 5 and is not divided.
Your sincerely,
Katarzyna Nowomiejska
Corresponding author
Round 2
Reviewer 1 Report (Previous Reviewer 1)
Line 120: Both eyes of each patient were tested.
As Armstrong wrote, ‘measurements obtained from the right and left eye of a subject are often correlated whereas many statistical tests assume observations in a sample are independent’.
Although LHON is an asymmetric disease, data from the right and left eyes of the same LHON patient are not independent but correlated. You cannot analyze them as independent data.
As Armstrong writes, authors should consider the following
(1) investigators should consider whether it is advantageous to collect data from both eyes
(2) if one eye is studied and both are eligible, then it should be chosen at random
(3) two-eye data can be analyzed incorporating eyes as a within-subjects factor
The authors want to analyze both eyes of LHON patients and of one NA-AION patient. Because these diseases are rare and if they include only one eye per patient, the sample size would be small. So the authors chose (1) instead of (2), and I understand that. However, if so, a statistical analysis should be performed that takes into account within-subject factors, which the authors have not done.
As I wrote in the previous review, only one eye per subject should be analyzed in both the patient and control groups, or statistical methods that account for within-subject bias should be used. However, if that is difficult, at least it should be stated as the limitation.
Taking the within-subject bias issue into account may probably not significantly change the analysis and will most likely not affect the results. However, this is an important issue. This issue concerns all statistical analyses in this paper, and if the authors do not properly address this issue, it is difficult for me to accept this paper. I have already pointed out this issue multiple times, but there has been no improvement.
Author Response
Dear Reviewer 1,
Thank you for your comments.
According to your suggestions, we have chosen one eye at random from each patient in each group and implemented updated results with tables and figures.
In the abstract it is now written (line 19-20):
12 eyes with chronic LHON and 10 eyes with chronic NA-AION and 8 NA-AION fellow eyes have been examined with OCT-A. Control group consisted of 20 eyes of 20 patients.
Instead of:
24 eyes of 12 patients with chronic LHON and 12 eyes of 10 patients with chronic NA-AION and 8 NA-AION fellow eyes have been examined with OCT-A. Control group consisted of 40 eyes of 20 patients.
In the abstract it is now written (line 22-30):
Results: There were significant differences in all sectors between groups in regard to the superficial vessel density and the inner and full thickness of the retina. Nasal sector of the macular superficial vessel density is affected more in LHON than in NA-AION, the same is with temporal sector of the retinal thickness. There were no significant differences between groups in the deep vessel plexus. There were no significant differences between vasculature of the inferior and superior hemifields of the macula in all groups and no correlation with the visual function. Conclusions: Superficial perfusion and structure of the macula assessed with OCT-A are affected both in the chronic LHON and NA-AION, but more in LHON eyes, especially in the nasal and temporal sectors.
Instead of:
Results: There were significant differences in the superficial vessel density between LHON and NA-AION eyes, apart from the nasal sector. The values of the whole as well as superior and inferior hemifields of the deep vessel plexus were significantly lower in NA-AION than in LHON. Inner and full thickness of the macula were significantly lower in LHON than in NA-AION in the inferior and temporal sectors. There were no significant differences between vasculature of the inferior and superior hemifields of the macula in all groups. Conclusions: Perfusion and structure of the macula are affected both in chronic LHON and NA-AION. OCT-A enables to detect differences in the deep vascular plexus and thickness of the retina in the chronic phase of these neuropathies.
In the methods section (line 76-79) it has been written:
12 eyes of 12 patients (11 males, 1 female; mean age 36 years) with molecularly confirmed LHON have been included in the study. All patients presented chronic phase of LHON according to Carelli [20] and Balducci [21] - more than 12 months from visual loss onset. Moreover, 10 eyes with the history of NA-AION onset in the past were also included (7 males, 3 females; mean age 60 years).
Instead of:
24 eyes of 12 patients (11 males, 1 female; mean age 36 years) with molecularly confirmed LHON have been included in the study. All patients presented chronic phase of LHON according to Carelli [20] and Balducci [21] - more than 12 months from visual loss onset. Moreover, 12 eyes with the history of NA-AION onset in the past were also included (2 patients with bilateral AION and 8 with monocular AION; 7 males, 3 females; mean age 60 years).
Line 98: The control group consisted of 20 eyes of 20 normal individuals in the similar age.
Instead of:
The control group consisted of 40 eyes of 20 normal individuals in the similar age.
In the statistical analysis section it has been written as follows:
Line 120-123: Both eyes of each patient have been examined, but only one eye was taken for analysis at random, as data from the right and left eyes of the same patient in binocular diseases are not independent but correlated.
Instead of:
Both eyes of each patient have been examined.
The following sentences have been deleted from the methods section: In case of AION patients each eye was taken into account as AION is a monocular disease. In case of LHON each eye was taken into analysis separately, as LHON is asymetrical disease and there is a systematic difference between the eyes. In the control group the mean value of both eyes was taken into account as the correlation between eyes was high (R=80-R=95).
The table 1 is as follows:
|
Parameter |
AION |
AION fellow eye |
LHON |
control<40 years |
control>40 years |
Statistical analysis |
|
|
||||||||||
|
Mean |
Median |
Standard deviation |
Mean |
Median |
Standard deviation |
Mean |
Median |
Standard deviation |
Mean |
Median |
Standard deviation |
Mean |
Median |
Standard deviation |
Kruskal-Wallis test |
Mann-Whitney test for LHON and NA-AION Z |
p |
|
|
whole superficial |
40.35 |
38.30 |
4.48 |
45.60 |
47.75 |
5.63 |
36.41 |
38.65 |
8.39 |
50.05 |
50.05 |
3.48 |
50.65 |
51.65 |
2.55 |
H=29.63, p<0,0001* |
-0.79 |
0.43 |
|
superior hemi superficial |
40.92 |
39.85 |
4.20 |
46.13 |
48.10 |
5.22 |
36.83 |
38.40 |
8.67 |
50.18 |
49.75 |
3.49 |
51.15 |
51.15 |
2.52 |
H=29.96, p<0,0001* |
-0.92 |
0.36 |
|
inferior hemi |
39.74 |
38.65 |
5.98 |
45.00 |
47.35 |
6.23 |
35.91 |
37.85 |
8.16 |
49.90 |
49.15 |
3.67 |
50.15 |
51.45 |
2.92 |
H=27,75, p<0,0001* |
-0.69 |
0.49 |
|
parafovea superficial |
42.67 |
42.90 |
5.82 |
46.23 |
47.10 |
8.24 |
37.19 |
39.80 |
8.71 |
52.42 |
52.75 |
3.69 |
52.92 |
53.10 |
2.14 |
H=28.46, p<0,0001* |
-1.62 |
0.11 |
|
superior |
42.13 |
44.75 |
9.55 |
45.55 |
47.95 |
12.26 |
38.35 |
39.65 |
7.59 |
53.83 |
53.95 |
3.01 |
54.37 |
54.35 |
1.17 |
H=27.36, p<0,0001* |
-1.68 |
0.09 |
|
inferior |
42.00 |
42.60 |
9.43 |
47.86 |
54.00 |
10.01 |
37.98 |
38.35 |
8.97 |
52.36 |
52.20 |
4.52 |
53.73 |
54.15 |
2.26 |
H=20.98, p=0,0001* |
-1.52 |
0.13 |
|
nasal |
42.63 |
42.65 |
4.90 |
44.48 |
44.00 |
7.13 |
33.98 |
39.10 |
11.37 |
51.15 |
51.90 |
5.33 |
52.22 |
52.50 |
3.39 |
H=31.68, p<0,0001* |
-2.80 |
0.01 |
|
temporal |
43.89 |
42.80 |
4.32 |
46.91 |
46.95 |
6.97 |
38.28 |
39.95 |
9.40 |
52.41 |
52.30 |
3.11 |
51.43 |
51.95 |
3.64 |
H=25.11, p<0,0001* |
-1.58 |
0.11 |
|
fovea |
11.75 |
13.15 |
6.97 |
16.96 |
16.40 |
7.27 |
14.75 |
14.15 |
3.71 |
20.37 |
21.90 |
5.49 |
21.38 |
22.10 |
2.80 |
H=15.78 p=0.003* |
0.88 |
0.38 |
The results section:
Instead of:
|
Parameter |
AION |
AION fellow eye |
LHON |
control<40 years |
control>40 years |
Statistical analysis |
|
|
||||||||||
|
Mean |
Median |
Standard deviation |
Mean |
Median |
Standard deviation |
Mean |
Median |
Standard deviation |
Mean |
Median |
Standard deviation |
Mean |
Median |
Standard deviation |
Kruskal-Wallis test |
Mann-Whitney test for LHON and NA-AION Z |
p |
|
|
whole superficial |
39.42 |
38.25 |
4.60 |
45.60 |
47.75 |
5.63 |
37.42 |
37.20 |
3.67 |
50.97 |
50.20 |
2.20 |
52.13 |
52.20 |
2.10 |
H=36.52 p<0.0001* |
0.00 |
1.00 |
|
superior hemi superficial |
40.07 |
38.65 |
4.31 |
46.13 |
48.10 |
5.22 |
37.77 |
37.45 |
3.98 |
51.09 |
50.80 |
2.32 |
52.48 |
52.44 |
1.48 |
H=36.48 p<0.0001* |
-0.41 |
0.68 |
|
inferior hemi |
38.73 |
37.90 |
5.94 |
45.00 |
47.35 |
6.23 |
37.16 |
37.35 |
3.38 |
50.81 |
50.05 |
2.21 |
51.66 |
51.95 |
2.76 |
H=31.46 p<0.0001* |
0.05 |
0.96 |
|
parafovea superficial |
41.65 |
41.20 |
6.00 |
46.23 |
47.10 |
8.24 |
39.46 |
40.05 |
3.57 |
53.69 |
53.65 |
2.36 |
54.65 |
55.15 |
2.28 |
H=33.30 p<0.0001* |
-1.12 |
0.26 |
|
superior |
41.30 |
43.55 |
9.13 |
45.55 |
47.95 |
12.26 |
39.25 |
40.40 |
3.83 |
54.87 |
54.60 |
2.55 |
56.47 |
56.60 |
2.12 |
H=32.67 p<0.0001* |
-1.33 |
0.18 |
|
inferior |
41.53 |
42.05 |
8.60 |
47.86 |
54.00 |
10.01 |
40.06 |
40.55 |
5.08 |
54.41 |
54.70 |
2.44 |
55.51 |
55.00 |
2.58 |
H=30.43 p<0.0001* |
-1.26 |
0.21 |
|
nasal |
40.91 |
41.45 |
7.13 |
44.48 |
44.00 |
7.13 |
39.09 |
39.70 |
4.49 |
52.53 |
51.90 |
2.77 |
54.07 |
53.80 |
2.61 |
H=34.80 p<0.0001* |
-1.71 |
0.09 |
|
temporal |
42.84 |
42.20 |
4.62 |
46.91 |
46.95 |
6.97 |
39.28 |
39.95 |
3.70 |
53.00 |
53.45 |
2.41 |
52.63 |
52.75 |
2.15 |
H=31.64 p<0.0001* |
-1.50 |
0.13 |
|
fovea |
11.75 |
13.15 |
6.97 |
16.96 |
16.40 |
7.27 |
14.75 |
14.15 |
3.71 |
20.37 |
21.90 |
5.49 |
21.38 |
22.10 |
2.80 |
H=15.78 p=0.003* |
0.88 |
0.38 |
Figure 1 is as follows:
Instead of:
It is now written (line132-133): There were significant differences (p<0.0001) of the superficial vessel density (%) between groups (Table 1, Figure 1), apart from the fovea (p=0.24).
Instead of:
There were significant differences of the superficial vessel density (%) between groups (Table 1, Figure 1).
The following sentence has been deleted:
The most decreased superficial vessel density of the fovea was observed in AION eyes.
It is now written (line 134-136):
However, there were no significant differences (p>0.05) between LHON and NA-AION eyes in macular superficial vessel density (table 1), apart from the nasal sector (p=0.01).
Instead of:
However, there were no significant differences between LHON and NA-AION eyes in macular superficial vessel density (table 1).
Table 2 is as follows:
|
Parameter |
AION |
AION fellow eye |
LHON |
control<40 years |
control>40 years |
Statistical analysis |
|
|
|
||||||||||||
|
Mean |
Median |
Standard deviation |
Mean |
Median |
Standard deviation |
Mean |
Median |
Standard deviation |
Mean |
Median |
Standard deviation |
Mean |
Median |
Standard deviation |
Kruskal-Wallis test |
Mann-Whitney test for LHON and NA-AION Z |
p |
||||
|
whole % deep |
46.15 |
46.50 |
5.87 |
45.24 |
46.05 |
6.84 |
47.63 |
52.10 |
13.54 |
51.57 |
51.10 |
6.36 |
51.33 |
52.60 |
2.74 |
H=8.06 p=0.15 |
1.45 |
0.15 |
|||
|
superior -hemi % deep |
47.48 |
48.65 |
6.37 |
47.23 |
48.20 |
6.57 |
48.94 |
53.65 |
13.11 |
52.46 |
52.30 |
6.40 |
52.48 |
53.30 |
3.05 |
H=7.37 p=0.19 |
1.42 |
0.16 |
|||
|
inferior-hemi % deep |
44.72 |
45.65 |
5.84 |
43.23 |
42.95 |
7.01 |
46.14 |
50.75 |
14.53 |
50.67 |
50.35 |
6.53 |
50.25 |
51.30 |
3.28 |
H=7.99 p=0.16 |
1.42 |
0.16 |
|||
|
parafovea deep % |
52.46 |
53.05 |
5.07 |
50.61 |
51.40 |
5.09 |
51.93 |
56.70 |
14.39 |
56.10 |
55.70 |
4.47 |
55.43 |
55.20 |
2.77 |
H=8.29 p=0.14 |
0.96 |
0.34 |
|||
|
superior |
52.17 |
53.15 |
5.90 |
50.66 |
51.15 |
5.33 |
53.53 |
57.90 |
11.62 |
55.39 |
55.05 |
5.16 |
54.32 |
54.20 |
2.21 |
H=8.09 p=0.15 |
1.42 |
0.16 |
|||
|
inferior |
50.05 |
50.35 |
7.64 |
47.38 |
48.75 |
6.39 |
51.72 |
56.60 |
14.64 |
54.73 |
54.05 |
5.40 |
54.90 |
55.45 |
2.67 |
H=9.71 p=0.08 |
1.38 |
0.17 |
|||
|
nasal |
53.05 |
53.55 |
4.47 |
52.50 |
53.65 |
4.20 |
50.11 |
55.40 |
16.48 |
57.05 |
57.25 |
4.02 |
56.33 |
56.40 |
4.09 |
H=7.44. p=0.19 |
0.56 |
0.58 |
|||
|
temporal |
54.57 |
56.20 |
4.88 |
51.95 |
53.40 |
5.51 |
52.48 |
56.50 |
15.09 |
57.29 |
56.40 |
4.07 |
56.17 |
55.15 |
3.10 |
H=7.67 p=0.17 |
0.63 |
0.53 |
|||
|
fovea |
30.20 |
29.90 |
6.26 |
33.23 |
35.70 |
11.73 |
31.28 |
32.90 |
8.26 |
36.44 |
37.00 |
9.15 |
39.33 |
39.30 |
6.18 |
H=7.46 p=0.19 |
0.82 |
0.41 |
|||
Instead of:
|
Parameter |
AION |
AION fellow eye |
LHON |
control<40 years |
control>40 years |
Statistical analysis |
|
|
||||||||||||
|
Mean |
Median |
Standard deviation |
Mean |
Median |
Standard deviation |
Mean |
Median |
Standard deviation |
Mean |
Median |
Standard deviation |
Mean |
Median |
Standard deviation |
Kruskal-Wallis test |
Mann-Whitney test for LHON and NA-AION Z |
|
|||
|
whole % deep |
45.46 |
46.00 |
5.99 |
45.24 |
46.05 |
6.84 |
50.07 |
51.45 |
6.24 |
52.59 |
53.40 |
4.11 |
55.90 |
55.55 |
3.52 |
H=17.07 p=0.002* |
2.34 |
|
||
|
superior -hemi % deep |
46.92 |
48.65 |
6.66 |
47.23 |
48.20 |
6.57 |
50.73 |
52.90 |
6.61 |
53.79 |
53.95 |
4.19 |
57.57 |
56.90 |
3.10 |
H=16.90 P=0.002* |
2.11 |
|
||
|
inferior-hemi % deep |
43.92 |
43.60 |
5.79 |
43.23 |
42.95 |
7.01 |
49.57 |
50.20 |
6.00 |
51.42 |
52.75 |
4.21 |
54.26 |
54.20 |
4.14 |
H=17.03 p=0.002* |
2.45 |
|
||
|
parafovea deep % |
51.55 |
53.05 |
6.64 |
50.61 |
51.40 |
5.09 |
55.56 |
56.75 |
4.96 |
56.27 |
56.05 |
3.14 |
58.60 |
58.05 |
2.67 |
H=13.17 p=0.01* |
1.26 |
|
||
|
superior |
51.17 |
53.15 |
7.39 |
50.66 |
51.15 |
5.33 |
55.63 |
56.95 |
5.70 |
55.79 |
55.90 |
3.31 |
58.05 |
57.55 |
2.94 |
H=10.60 p=0.03* |
1.60 |
|
||
|
inferior |
50.37 |
50.35 |
7.31 |
47.38 |
48.75 |
6.39 |
55.74 |
56.70 |
5.72 |
55.37 |
55.35 |
3.67 |
58.32 |
58.25 |
3.09 |
H=16.13 p=0.03* |
1.71 |
|
||
|
nasal |
50.57 |
53.55 |
10.48 |
52.50 |
53.65 |
4.20 |
55.48 |
55.95 |
4.97 |
56.67 |
56.55 |
3.34 |
59.14 |
57.85 |
2.23 |
H=13.31 p=0.01* |
1.12 |
|
||
|
temporal |
54.07 |
56.20 |
5.29 |
51.95 |
53.40 |
5.51 |
55.42 |
55.95 |
4.56 |
57.23 |
56.85 |
2.79 |
58.90 |
58.90 |
2.58 |
H=10.62 p=0.03* |
0.47 |
|
||
|
fovea |
27.96 |
28.40 |
7.80 |
33.23 |
35.70 |
11.73 |
31.28 |
32.25 |
5.97 |
37.99 |
39.30 |
5.67 |
40.45 |
40.10 |
2.40 |
H=15.28 p=0.004* |
1.08 |
|
||
Figure 2 is as follows:
Instead of:
In the results section it is now written (line 149-150):
There were no significant differences (p<0.05) between groups in regard to deep vessel density (Table 2, Figure 2).
Instead of:
There were significant differences between groups in regard to deep vessel density (Table 2, Figure 2).
The following sentences have been removed:
The lowest values of the whole and sectoral deep vessel density were were obtained in AION and AION fellow eyes. The most decreased deep vessel density of the fovea was observed in AION eyes.
It is written (line 150-151):
Moreover, there were no significant differences between LHON and NA-AION group (table 2).
Instead of:
However, significant differences between LHON and NA-AION have been found for the whole deep vessel density, superior hemifield and inferior hemifield (table 2).
Table 3 is as follows:
|
Parameter |
AION |
AION fellow eye |
LHON |
control<40 years |
control>40 years |
Statistical analysis |
|
|
||||||||||
|
Mean |
Median |
Standard deviation |
Mean |
Median |
Standard deviation |
Mean |
Median |
Standard deviation |
Mean |
Median |
Standard deviation |
Mean |
Median |
Standard deviation |
Kruskal-Wallis test |
Mann-Whitney test for LHON and NA-AION Z |
p |
|
|
superior |
71.90 |
65.50 |
26.09 |
91.00 |
102.0 |
31.44 |
58.92 |
59.50 |
6.02 |
116.00 |
115.50 |
6.85 |
117.17 |
117.00 |
3.66 |
H=28.22 p<0.0001* |
-1.15 |
0.25 |
|
inferior |
76.20 |
75.50 |
27.58 |
96.13 |
100.5 |
28.19 |
59.50 |
59.00 |
6.75 |
114.93 |
116.00 |
7.17 |
116.17 |
117.00 |
3.06 |
H=29.89 p<0.0001* |
-1.85 |
0.06 |
|
nasal |
71.80 |
70.00 |
23.86 |
88.50 |
99.5 |
27.89 |
59.00 |
56.00 |
11.11 |
106.14 |
115.50 |
28.40 |
110.83 |
110.50 |
3.31 |
H=24.77 p=0.0001* |
-0.76 |
0.45 |
|
temporal |
71.40 |
70.00 |
17.84 |
91.63 |
95.0 |
13.24 |
55.83 |
57.00 |
6.66 |
104.36 |
105.00 |
5.34 |
106.33 |
105.00 |
5.75 |
H=34.91 p<0.0001* |
-2.18 |
0.03* |
|
fovea |
35.50 |
37.50 |
13.10 |
44.63 |
45.5 |
13.21 |
30.92 |
30.50 |
2.94 |
52.29 |
54.00 |
11.84 |
57.00 |
57.00 |
6.45 |
H=24.55 p=0.0001* |
-0.56 |
0.58 |
Instead of:
|
Parameter |
AION |
AION fellow eye |
LHON |
control<40 years |
control>40 years |
Statistical analysis |
|
|
||||||||||
|
Mean |
Median |
Standard deviation |
Mean |
Median |
Standard deviation |
Mean |
Median |
Standard deviation |
Mean |
Median |
Standard deviation |
Mean |
Median |
Standard deviation |
Kruskal-Wallis test |
Mann-Whitney test for LHON and NA-AION Z |
p |
|
|
superior |
69.92 |
60.00 |
24.05 |
91.00 |
102.0 |
31.44 |
56.08 |
55.50 |
6.04 |
117.20 |
119.00 |
4.81 |
116.80 |
118.50 |
4.83 |
H=30,89 p<0.0001* |
-1.91 |
0.06 |
|
inferior |
73.00 |
69.50 |
26.05 |
96.13 |
100.5 |
28.19 |
56.17 |
56.50 |
5.54 |
116.20 |
118.00 |
5.26 |
116.40 |
116.00 |
3.85 |
H=32,53 p<0.0001* |
-2.09 |
0.04* |
|
nasal |
69.25 |
59.50 |
22.39 |
88.50 |
99.5 |
27.89 |
55.25 |
54.50 |
4.54 |
108.63 |
116.50 |
20.65 |
112.80 |
113.50 |
5.12 |
H=29,59 p<0.0001* |
-1.05 |
0.30 |
|
temporal |
68.83 |
67.00 |
17.26 |
91.63 |
95.0 |
13.24 |
52.67 |
52.50 |
5.73 |
105.47 |
106.00 |
3.94 |
105.80 |
105.00 |
4.16 |
H=37,88 p<0.0001* |
-2.56 |
0.01* |
|
fovea |
33.75 |
30.00 |
12.56 |
44.63 |
45.5 |
13.21 |
29.33 |
29.00 |
2.81 |
55.27 |
58.50 |
8.19 |
55.80 |
55.00 |
2.84 |
H=29,86 p<0.0001* |
-0.05 |
0.96 |
Figue 3 is a s follows:
Instead of:
It is written as follows (line 164-167):
There were significant differences (p<0.0001)of all values in the inner thickness of the retina between groups (Table 3). There was significant difference (p=0.03) between LHON and NA-AION eyes in regard to temporal sector of the inner thickness of the macula (Figure 3).
Instead of:
There were significant differences of all values in the inner thickness of the retina between groups (Table 3). There were significant differences between LHON and NA-AION eyes in regard to inferior and temporal sectors of the inner thickness of the macula (Figure 3).
The following sentence has been deleted:
The lowest values of the sectoral inner thicknesss and fovea were obtained in LHON eyes.
Table 4 is as follows:
|
Parameter |
AION |
AION fellow eye |
LHON |
control<40 years |
control>40 years |
Statistical analysis |
|
|
|||||||||||||
|
Mean |
Median |
Standard deviation |
Mean |
Median |
Standard deviation |
Mean |
Median |
Standard deviation |
Mean |
Median |
Standard deviation |
Mean |
Median |
Standard deviation |
Kruskall-Wallis test |
Mann-Whitney test for LHON and NA-AION Z |
p |
|
|||
|
superior |
292.60 |
295.50 |
24.71 |
335.00 |
337.00 |
14.15 |
283.08 |
282.50 |
14.02 |
338.00 |
336.00 |
13.20 |
338.67 |
336.50 |
7.99 |
H=33.29. p<0.0001* |
-1.12 |
0.26 |
|
||
|
inferior |
309.60 |
307.00 |
24.02 |
328.50 |
326.00 |
13.24 |
280.25 |
283.00 |
14.64 |
333.79 |
338.50 |
11.17 |
337.33 |
334.50 |
6.38 |
H=30.97. p<0.0001* |
-2.90 |
0.004* |
|
||
|
nasal |
302.70 |
307.50 |
28.86 |
332.38 |
333.50 |
12.84 |
277.67 |
281.50 |
20.79 |
336.43 |
338.00 |
13.21 |
339.50 |
337.50 |
6.66 |
H=30.59. p<0.0001* |
-1.95 |
0.05 |
|
||
|
temporal |
298.60 |
297.00 |
23.43 |
323.13 |
324.50 |
12.52 |
276.25 |
278.00 |
12.61 |
323.79 |
325.50 |
10.34 |
324.50 |
320.50 |
7.97 |
H=27.76. p<0.0001* |
-2.01 |
0.04* |
|
||
|
fovea |
253.20 |
250.00 |
22.69 |
269.75 |
273.00 |
31.44 |
229.83 |
230.00 |
27.68 |
253.14 |
254.00 |
22.78 |
274.83 |
272.50 |
9.75 |
H=14.48. p=0.006* |
-1.85 |
0.06 |
|
||
|
FAZ |
0.30 |
0.30 |
0.08 |
0.28 |
0.29 |
0.08 |
0.28 |
0.28 |
0.07 |
0.29 |
0.30 |
0.11 |
0.23 |
0.22 |
0.09 |
H=3.03. p=0.55 |
-0.69 |
0.49 |
|
||
Instead of:
|
Parameter |
AION |
AION fellow eye |
LHON |
control<40 years |
control>40 years |
Statistical analysis |
|
|
||||||||||
|
Mean |
Median |
Standard deviation |
Mean |
Median |
Standard deviation |
Mean |
Median |
Standard deviation |
Mean |
Median |
Standard deviation |
Mean |
Median |
Standard deviation |
Kruskal-Wallis test |
Mann-Whitney test for LHON and NA-AION Z |
p |
|
|
superior |
71.90 |
65.50 |
26.09 |
91.00 |
102.0 |
31.44 |
58.92 |
59.50 |
6.02 |
116.00 |
115.50 |
6.85 |
117.17 |
117.00 |
3.66 |
H=28.22 p<0.0001* |
-1.15 |
0.25 |
|
inferior |
76.20 |
75.50 |
27.58 |
96.13 |
100.5 |
28.19 |
59.50 |
59.00 |
6.75 |
114.93 |
116.00 |
7.17 |
116.17 |
117.00 |
3.06 |
H=29.89 p<0.0001* |
-1.85 |
0.06 |
|
nasal |
71.80 |
70.00 |
23.86 |
88.50 |
99.5 |
27.89 |
59.00 |
56.00 |
11.11 |
106.14 |
115.50 |
28.40 |
110.83 |
110.50 |
3.31 |
H=24.77 p=0.0001* |
-0.76 |
0.45 |
|
temporal |
71.40 |
70.00 |
17.84 |
91.63 |
95.0 |
13.24 |
55.83 |
57.00 |
6.66 |
104.36 |
105.00 |
5.34 |
106.33 |
105.00 |
5.75 |
H=34.91 p<0.0001* |
-2.18 |
0.03* |
|
fovea |
35.50 |
37.50 |
13.10 |
44.63 |
45.5 |
13.21 |
30.92 |
30.50 |
2.94 |
52.29 |
54.00 |
11.84 |
57.00 |
57.00 |
6.45 |
H=24.55 p=0.0001* |
-0.56 |
0.58 |
Figure 4 is as follows:
Instead of:
Table 5 is as follows:
|
parameter |
NA-AION |
AION fellow eyes |
LHON |
control group |
||||||||
|
Mean |
Median |
Standard deviation |
Mean |
Median |
Standard deviation |
Mean |
Median |
Standard deviation |
Mean |
Median |
Standard deviation |
|
|
Vessel density superficial
|
Z=1.10; p=0.92 |
Z=0.85; p=0.40 |
Z=0.31; p=0.75 |
Z=2.35; p=0.02* |
||||||||
|
superior |
42.13 |
44.75 |
9.55 |
45.55 |
42.13 |
44.75 |
9.55 |
45.55 |
42.13 |
44.75 |
9.55 |
45.55 |
|
inferior |
42.00 |
42.60 |
9.43 |
47.86 |
42.00 |
42.60 |
9.43 |
47.86 |
42.00 |
42.60 |
9.43 |
47.86 |
|
Absolute value of the difference between hemifields |
3.35 |
2.70 |
2.43 |
5.56 |
3.35 |
2.70 |
2.43 |
5.56 |
3.35 |
2.70 |
2.43 |
5.56 |
|
Statistical analysis |
H=5..42. p=0.14 |
|||||||||||
|
Vessel density deep
|
Z=1.48; p=0.14 |
Z=2.10; p=0.02* |
Z=1.84; p=0.07 |
Z=0.50; p=0.61 |
||||||||
|
superior |
52.17 |
53.15 |
5.90 |
50.66 |
52.17 |
53.15 |
5.90 |
50.66 |
52.17 |
53.15 |
5.90 |
50.66 |
|
inferior |
50.05 |
50.35 |
7.64 |
47.38 |
50.05 |
50.35 |
7.64 |
47.38 |
50.05 |
50.35 |
7.64 |
47.38 |
|
Absolute value of the difference between hemifields |
4.70 |
2.90 |
4.62 |
3.44 |
4.70 |
2.90 |
4.62 |
3.44 |
4.70 |
2.90 |
4.62 |
3.44 |
|
Statistical analysis |
H=8.96. p=0.03*; no differeneces between groups
|
|||||||||||
|
Inner thickness
|
Z=0.95; p=0.34 |
Z=0.42; p=0.67 |
Z=0.67; p=0.50 |
Z=1.41; p=0.16 |
||||||||
|
superior |
71.90 |
65.50 |
26.09 |
91.00 |
71.90 |
65.50 |
26.09 |
91.00 |
71.90 |
65.50 |
26.09 |
91.00 |
|
inferior |
76.20 |
75.50 |
27.58 |
96.13 |
76.20 |
75.50 |
27.58 |
96.13 |
76.20 |
75.50 |
27.58 |
96.13 |
|
Absolute value of the difference between hemifields |
11.70 |
6.50 |
11.51 |
9.88 |
11.70 |
6.50 |
11.51 |
9.88 |
11.70 |
6.50 |
11.51 |
9.88 |
|
Statistical analysis |
|
H=10.13. p=0.02*; no differeneces between groups
|
||||||||||
|
Full thickness
|
Z=1.17; p=0.24 |
Z=1.86; p=0.06 |
Z=1.83; p=0.07 |
Z=2.24; p=0.03* |
||||||||
|
superior |
292.60 |
295.50 |
24.71 |
335.00 |
292.60 |
295.50 |
24.71 |
335.00 |
292.60 |
295.50 |
24.71 |
335.00 |
|
inferior |
309.60 |
307.00 |
24.02 |
328.50 |
309.60 |
307.00 |
24.02 |
328.50 |
309.60 |
307.00 |
24.02 |
328.50 |
|
Absolute value of the difference between hemifields |
20.60 |
6.50 |
32.70 |
7.50 |
20.60 |
6.50 |
32.70 |
7.50 |
20.60 |
6.50 |
32.70 |
7.50 |
|
Statistical analysis |
H=3.68. p=0.30 |
|||||||||||
Instead of:
|
parameter |
NA-AION |
AION fellow eyes |
LHON |
control group |
||||||||
|
Mean |
Median |
Standard deviation |
Mean |
Median |
Standard deviation |
Mean |
Median |
Standard deviation |
Mean |
Median |
Standard deviation |
|
|
Vessel density superficial
|
Z=0.09. p=0.93 |
Z=-0.26; p=0.79 |
Z=0.40; p=0.69 |
Z=0.50; p=0.62 |
||||||||
|
superior |
41.30 |
43.55 |
9.13 |
45.55 |
47.95 |
12.26 |
39.55 |
40.95 |
5.98 |
55.27 |
54.73 |
2.50 |
|
inferior |
41.53 |
42.05 |
8.60 |
47.86 |
54.00 |
10.01 |
39.59 |
39.55 |
7.34 |
54.68 |
54.75 |
2.46 |
|
Absolute value of the difference between hemifields |
3.67 |
2.95 |
2.48 |
5.56 |
3.40 |
6.30 |
4.28 |
3.75 |
3.44 |
1.14 |
1.10 |
0.88 |
|
Statistical analysis |
H=17.12, p=0.0007* NA-AION- control group Z=3.05, p=0.01*; LHON-control group Z=3.81, p=0.0008* |
|||||||||||
|
Vessel density deep
|
Z=-0.87; p=0.39 |
Z=-2.42; p=0.02* |
Z=0.00; p=1.00 |
Z=0.18; p=0.86 |
||||||||
|
superior |
51.17 |
53.15 |
7.39 |
50.66 |
51.15 |
5.33 |
53.84 |
56.20 |
9.29 |
56.35 |
56.60 |
3.31 |
|
inferior |
50.37 |
50.35 |
7.31 |
47.38 |
48.75 |
6.39 |
53.21 |
55.95 |
11.33 |
56.11 |
55.80 |
3.70 |
|
Absolute value of the difference between hemifields |
4.88 |
2.90 |
4.77 |
3.44 |
3.40 |
2.79 |
2.49 |
1.80 |
2.45 |
0.76 |
0.48 |
0.71 |
|
Statistical analysis |
H=16.11, p=0.001*; NA-AION- control group Z=3.02, p=0.02*; LHON- control group Z=3.28, p=0.006*; AION fellow eyes- control group Z=2.94, p=0.02* |
|||||||||||
|
Inner thickness
|
Z=-0.29; p=077 |
Z=0.00; p=1.00 |
Z=-0.11; p=0.92 |
Z=0.51; p=0.61 |
||||||||
|
superior |
69.92 |
60.00 |
24.05 |
91.000 |
102.00 |
31.44 |
57.64 |
57.50 |
5.59 |
117.10 |
118.75 |
4.69 |
|
inferior |
73.00 |
69.50 |
26.05 |
96.125 |
100.50 |
28.19 |
57.86 |
58.00 |
5.78 |
116.25 |
117.50 |
4.85 |
|
Absolute value of the difference between hemifields |
10.25 |
5.50 |
10.95 |
9.88 |
5.50 |
10.43 |
2.05 |
1.50 |
1.73 |
1.60 |
1.00 |
1.76 |
|
Statistical analysis |
|
H=17.24, p=0.0006*; NA-AION- control group Z=3.11, p=0.01*; LHON- AION fellow eyes- Z=2.65, p=0.05*; AION fellow eyes- control group Z=3.27, p=0.007* |
||||||||||
|
Full thickness
|
Z=-1.04; p=0.30 |
Z=0.99; p=0.32 |
Z=0.56; p=0.57 |
Z=1.14; p=0.25 |
||||||||
|
superior |
291.67 |
292.50 |
22.66 |
335.00 |
337.00 |
14.15 |
284.82 |
283.00 |
12.60 |
336.95 |
338.75 |
8.72 |
|
inferior |
304.33 |
299.00 |
25.33 |
328.50 |
326.00 |
13.24 |
282.18 |
285.00 |
11.83 |
334.20 |
336.00 |
8.38 |
|
Absolute value of the difference between hemifields |
18.67 |
7.00 |
29.95 |
7.50 |
4.50 |
8.18 |
4.73 |
3.50 |
3.82 |
4.00 |
3.00 |
2.62 |
|
Statistical analysis |
H=4.87, p=0.19 |
|||||||||||
It is now written (line 202-205):
Statistical analysis showed that taking into account the absolute values of differences between superficial and deep vessel density of the superior and inferior hemifields there were no significant differences found between disease groups and control group, there were also no significant differences in regard to the full thickness and inner thickness of the retina (p>0.05) (Table 5).
Instead of:
Statistical analysis showed that taking into account the absolute values of differences between superficial and deep vessel density of the superior and inferior hemifields there were significant differences found between disease groups and control group, there were also no significant differences in regard to the full thickness and inner thickness of the retina (p>0.05) (Table 5).
It is now written (218-224):
Median visual acuity in the LHON group was 1.1 logMAR (SD ±0.63), 0.3 logMAR (SD ±0.61) in NA-AION group and 0.0 logMAR (SD ±0.06) in AION fellow eyes (p=0.001). Median visual field index was in 65.5% in LHON eyes (SD ±29.18%), 79% (SD±33.45%) in NA-AION eyes and 99% (SD±2.45%) in AION fellow eyes (p=0.02). Median MD value was -11.62 dB (SD±8.97dB) in LHON eyes, -8.76dB (SD±9.92dB) in NA-AION eyes and -1.54dB (SD±1.25dB) in AION fellow eyes (p=0.06). Median PSD value was 8.68dB (SD ±3.18dB) in LHON eyes, 8.95dB (SD ±4.25dB) in NA--AION eyes and 2.3 dB (SD±0.95dB) in AION fellow eyes (p=0.01).
Instead of:
There were significant differences in regard to the visual acuity and the visual field parameters between groups. Median visual acuity in the LHON group was 1.0 logMAR (SD ±0.6), 0.35 logMAR (SD ±0.56) in NA-AION group and 0.0 logMAR (SD ±0.06) in AION fellow eyes (p=0.0004). Median visual field index was in 69.5% in LHON eyes (SD ±28.19%), 79% (SD±30.39%) in NA-AION eyes and 99% (SD±2.45%) in AION fellow eyes (p=0.002). Median MD value was -9.875 dB (SD±8.49dB) in LHON eyes, -8.760dB (SD±9.22dB) in NA-AION eyes and -1.54dB (SD±1.25dB) in AION fellow eyes (p=0.001). Median PSD value was 8.610dB (SD ±3.2dB) in LHON eyes, 8.945dB (SD ±3.88dB) in NA--AION eyes and 2.3 dB (SD±0.95dB) in AION fellow eyes (p=0.005).
In the discussion chapter it is now written (274-276):
The major finding is that the vascular and structural alterations of the macula are present both in chronic LHON and NA-AION, although the vascular changes are more pronounced in LHON.
Instead of:
The major finding is that the vascular and structural alterations of the macula are present both in chronic LHON and NA-AION.
The following sentence has been deleted:
Whole deep vessel density is decreased in NA-AION eyes more than in eyes with previous episode of LHON.
It is written (line 277-279):
Full and inner thickness in the temporal sector of the macula are decreased more in chronic LHON than in chronic NA-AION.
Instead of:
Full and inner thickness in the inferior and temporal sectors of the macula are decreased more in chronic LHON than in chronic NA-AION.
It is now written (line 350-353):
Thinning of the retinal thickness is more pronounced in chronic LHON eyes (in the temporal sector) and reduction of the superficial vascular plexus is more pronounced in chronic NA-AION eyes.
Instead of:
Thinning of the retinal thickness is more pronounced in chronic LHON eyes (in the inferior and temporal sectors) and reduced deep vascular plexus is more pronounced in chronic NA-AION eyes.
It is now written (line 393-395):
Our study has shown that there vasculature of the macula is decreased both in LHON and in NA-AION eyes in a chronic phase of the diseases with some differences, especially in regard to superficial vascular plexus.
Instead of:
Our study has shown that there vasculature of the macula is decreased both in LHON and in NA-AION eyes in a chronic phase of the diseases with some differences, especially in regard to deep vascular plexus.
Your sincerely,
Katarzyna Nowomiejska
Corresponding author
This manuscript is a resubmission of an earlier submission. The following is a list of the peer review reports and author responses from that submission.
Round 1
Reviewer 1 Report
1. Introduction
(Line 55) The authors need to state more clearly why they compared LHON and NA-AION.
2.1 Study type (Line 61)
This study is a cross-sectional study and is not appropriate to be considered prospective.
2.2 Patient characteristics
Is the number of patients with monocular AION 8 or 10?
2.4 Statistical analysis (Line 107-108)
“Both eyes of each patient were tested, and the calculated means of the eye results were included in the statistical analysis.”
What about patients with unilateral NA-AION?
Did the authors also average the results of both eyes of monocular NA-AION patients?
Both LOHN and binocular NA-AION patients would have had different degrees of impairment in the left and right eye. Was there a difference in the degree of inter-eye difference between LHON and NA-AION patients? Averaging the results of both eyes may make it difficult to assess the topographic relationship between blood flow and retinal damage. I think it is better to include only one eye per patient rather than averaging both eyes’ results.
3. Results
The authors did not indicate the ophthalmologic findings of the subjects. They need to present the results of visual acuity and visual field testing and provide results on whether there were differences between LHON and NA-AION patients. In NA-AION, it would be better to have information on which hemifield (or quadrant) was damaged.
If data from both eyes are to be averaged and analyzed, data from each eye should also be shown.
The authors used the Kruskal-Wallis test for between-group comparisons; although differences between the four groups were shown, results of direct comparisons between LHON and NA-AION were not presented. The difference, if any, between LHON and NA-AION needs to be more clearly indicated.
As Figure 5 shows, LHON shows an overall decrease in macular blood flow and inner retinal thickness, whereas NA-AION shows differences in blood flow and inner retinal thickness in the upper and lower hemi-fields. These differences between LHON and NA-AION need to be analyzed in more detail.
4. Discussion
(Line 243-244) “The present study adds additional knowledge to the pathogenesis and natural history of LHON and NA-AION. “
I think it would be better to elaborate on this point.
5. Conclusion
(Line 258-259) “Our study has shown that there is more decreased vasculature of the macula in LHON eyes than in NA-AION eyes in a chronic phase of the disease. “
Again, it is not indicated whether there was a statistical difference in these parameters between LHON and NA-AION. The authors noted that the lowest values of the superficial vessel density were obtained in LHON eyes and the lowest values of the deep vessel density were obtained in AION and AION fellow eyes. The meaning of the sentence in Line 258-259 is not clear.
6. References
No. 6 and 23 seem to be the same paper.
Author Response
Reviewer 1
- Introduction
(Line 55) The authors need to state more clearly why they compared LHON and NA-AION.
In the introduction section (line 74-79) it has been added:
Both LHON and NA-AION share the mechanism of vascular abnormalities of the optic nerve in the acute phase, leading to profound central visual loss, although there is different pathogenesis of these entities. In the chronic phase it is difficult to distinguish between these neuropathies, as they both lead to the optic disc pallor. The chronic phase alterations of the macula in chronic LHON and NA-AION have not been investigated deeply so far.
The following reference has been added:
Asanad, S., Meer, E., Fantini, M., Borrelli, E., & Sadun, A. A. Leber's hereditary optic neuropathy: Shifting our attention to the macula. American journal of ophthalmology case reports. 2018, 13, 13–15.
2.1 Study type (Line 61)
This study is a cross-sectional study and is not appropriate to be considered prospective.
In the methods section (line 92) it is now written: This is a cross-sectional study.
2.2 Patient characteristics
Is the number of patients with monocular AION 8 or 10?
In the methods section (line 101-103) it is now written:
Moreover, 12 eyes with the history of NA-AION onset in the past were also included (2 patients with bilateral AION and 8 with monocular AION; 7 males, 3 females; mean age 60 years).
Instead of:
Moreover, 12 eyes with the history of NA-AION onset in the past were also included (2 patients with bilateral AION and 10 with monocular AION; 9 males, 3 females; mean age 60 years).
2.4 Statistical analysis (Line 107-108)
“Both eyes of each patient were tested, and the calculated means of the eye results were included in the statistical analysis.”
What about patients with unilateral NA-AION?
In the abstract (line 19-21) it is now written: 24 eyes of 12 patients with chronic LHON and 12 eyes of 10 patients with previous episode of visual loss due to NA-AION and 8 NA-AION fellow eyes have been examined with OCT-A. Control group consisted of 40 eyes of 20 patients.
In the methods section (line 98-99) it has been written:
24 eyes of 12 patients (11 males, 1 female; mean age 36 years) with molecularly confirmed LHON have been included in the study.
And later (line 136-138): The control group consisted of 40 eyes of 20 normal individuals in the similar age.
Did the authors also average the results of both eyes of monocular NA-AION patients?
Both LOHN and binocular NA-AION patients would have had different degrees of impairment in the left and right eye. Was there a difference in the degree of inter-eye difference between LHON and NA-AION patients? Averaging the results of both eyes may make it difficult to assess the topographic relationship between blood flow and retinal damage. I think it is better to include only one eye per patient rather than averaging both eyes’ results.
In the statistical analysis section (line 158-161) it is now written: Both eyes of each patient were tested, in case of AION, fellow AION eyes and LHON each eye was taken into analysis separately and for the control group the calculated means of both eye results were included in the statistical analysis.
- Results
The authors did not indicate the ophthalmologic findings of the subjects. They need to present the results of visual acuity and visual field testing and provide results on whether there were differences between LHON and NA-AION patients.
In the result section (line 241-250) it has been added:
3.6.Visual function results.
There were significant differences in regard to the visual acuity and the visual field parameters between groups. Median visual acuity in the LHON group was 1.0 logMAR (SD ±0.6), 0.35 logMAR (SD ±0.56) in NA-AION group and 0.0 logMAR (SD ±0.06) in AION fellow eyes (p=0,0004). Median visual field index was in 69.5% in LHON eyes (SD ±28.19%), 79% (SD ±30.39%) in NA-AION eyes and 99% (SD ±2.45%) in AION fellow eyes (p=0.002). Median MD value was -9.875 dB (SD ±8.49dB) in LHON eyes, -8.760dB (SD ±9.22dB) in AION eyes and -1.54dB (SD ±1.25dB) in NA-AION fellow eyes (p=0.001). Median PSD value was 8.610dB (SD ±3.2dB) in LHON eyes, 8.945dB (SD ±3.88dB) in NA-AION eyes and 2.3 dB (SD ±0.95dB) in AION fellow eyes (p=0,005). Additional statistical analysis revealed statistically significant differences between AION eyes and AION fellow eyes and between LHON eyes and AION fellow eyes in regard to all previously mentioned visual function parameters, but no differences between LHON and NA-AION.
In NA-AION, it would be better to have information on which hemifield (or quadrant) was damaged.
In the results section (line 252-253) it has been added:
Qualitative assessment of the visual field results revealed visual field defects in the upper hemifield in 4 eyes, in the lower hemifield in 5 eyes and in the lower quadrant in 3 eyes.
If data from both eyes are to be averaged and analyzed, data from each eye should also be shown.
In the statistical analysis section (line 158-161) it is now written: Both eyes of each patient were tested, in case of AION, fellow AION eyes and LHON each eye was taken into analysis separately and for the control group the calculated means of both eye results were included in the statistical analysis.
The authors used the Kruskal-Wallis test for between-group comparisons; although differences between the four groups were shown, results of direct comparisons between LHON and NA-AION were not presented. The difference, if any, between LHON and NA-AION needs to be more clearly indicated.
In the methods section (line 163-166) it has been added:
Mann-Whitney test has been used for comparisons of two independent groups (LHON and NA-AION).
In tables 1 to 4 additional column with results of Mann-Whitney test have been added.
Line 173-174: In the results section it has been added: However, there were no significant differences between LHON and NA-AION eyes in macular superfcial vessel density (table 1).
Line 188-190: However, significant differences between LHON and NA-AION have been found for the whole deep vessel density, superior hemifield and inferior hemifield (table 2).
Line 203-204: There were significant differences between LHON and NA-AION eyes in regard to inferior and temporal sectors of the inner thickness of the macula (table 3).
Line218-220: There were significant differences between LHON and NA-AION eyes in regard to inferior and temporal sectors of the full thickness of the macula (table 4).
In the abstract it is written:
Line 23-29: There were no significant differences between LHON and NA-AION eyes in macular superficial vessel density. The values of the whole as well as superior and inferior hemifields of the deep vessel plexus were significantly lower in NA-AION than in LHON eyes. Inner and full thickness of the macula were significantly lower in LHON than in NA-AION eyes only in inferior and temporal sectors.
Instead of:
All macular vessel density values (superficial and deep vessel density), as well as inner and full retinal thickness were statistically decreased in eyes with LHON, NA-AION and NA-AION fellow eyes in comparison to control groups. The lowest values of the whole and sectoral superficial vessel density, as well as sectoral full and inner thickness and fovea and were obtained in LHON eyes. The lowest values of the whole and sectoral deep vessel density were obtained in AION eyes.
In the discussion (line 275-279) chapter it is written:
The major finding is that the vascular and structural alterations of the macula are present both in chronic LHON and NA-AION. Whole deep vessel density is decreased in NA-AION eyes more than in eyes with previous episode of LHON. Full and inner thickness in the inferior and temporal secotors of the macula are decreased more in chronic LHON than in chronic NA-AION.
Instead of:
The major finding is that the vessel density is decreased in LHON eyes more than in eyes with previous NA-AION episode.
As Figure 5 shows, LHON shows an overall decrease in macular blood flow and inner retinal thickness, whereas NA-AION shows differences in blood flow and inner retinal thickness in the upper and lower hemi-fields. These differences between LHON and NA-AION need to be analyzed in more detail.
In the methods section (line 148-150) it has been added: All sectors of the macula have been analysed (inferior, superior, nasal, temporal), moreover, comparison between superior and inferior hemifields of the macula has been done in all the groups.
In the results section (line 232-235) it has been added:
3.5 Comparison of superior and inferior hemifields of the macula.
Statistical analysis showed no significant differences between superior and inferior hemifields of the macula in all groups in regard to superficial and deep vessel density, as well as inner thickness and full thickness of the macula (table 5).
Additionally, table 5 has been added:
Table 5. Comparison of the superior and inferior hemifields of the macula in the superficial and deep vessel density, as well as inner thickness and full thickness of the macula.
In the abstract (line 27-28) it has been added:
There are no significant differences between vasculature of the inferior and superior hemifields of the macula in all groups.
In the discussion chapter (line 362-365) it has been added:
Interestingely, there were no differences between upper and lower hemifields both in chronic NA-AION and LHON. It is already known that in the acute phase of NA-AION there is vascular density decrease with visual field defects and RNFL loss matching [32].
The following reference has been added:
Pierro, L., Arrigo, A., Aragona, E., Cavalleri, M., & Bandello, F. Vessel Density and Vessel Tortuosity Quantitative Analysis of Arteritic and Non-arteritic Anterior Ischemic Optic Neuropathies: An Optical Coherence Tomography Angiography Study. Journal of clinical medicine. 2020, 9(4), 1094.
As statistical analysis revealed no significant differences between hemifields, in figure 5 a picture of the macula of patient with NA-AION has been replaced by more representative with no differences in the superior and inferior hemifield of the macula.
Instead of:
- Discussion
(Line 243-244) “The present study adds additional knowledge to the pathogenesis and natural history of LHON and NA-AION. “
I think it would be better to elaborate on this point.
In the discussion chapter (line 347-380) it has been added:
It has been already shown that ganglion cell layer and inner plexiform layer are acutely unaffected in NA-AION after one month from the episode of the visual loss [30]. Thinning of ganglion cell layer develops within 1 to 2 months of onset, which is prior to RNFL swelling resolution. Knowing that approximately 50% of the retinal ganglion cells are located in the human macula region and the central 10° of the visual field is affected in NA-AION, exploring the macula region seems to be useful for detecting structural loss. Interestingly, in other optic neuropathy – optic neuritis – there is thinning of the retina ganglion cell layer plus inner plexiform layer at one-two months period after episode of optic neuritis [31].
Both LHON and NA-AION lead to the optic disc atrophy in the chronic stage. Our study showed that in the chronic phase of both LHON and NA-AION there are generally similar changes in the macular vasculature and retinal thickness, apart from some sectors, different from the controls. Thinning of the retinal thickness and reduced vascular plexus are probably due to the reduction in the overall metabolic rate of the retina in the chronic phase of these optic neuropathies.
Interestingely, there were no differences between upper and lower hemifields both in chronic NA-AION and LHON. It is already known that in the acute phase of NA-AION there is vascular density decrease with visual field defects and RNFL loss matching [32].
Fard and colleagues [33] compared the results of OCT-A in the chronic stage of two neuropathies: demyelinating optic neuritis and AION. They concluded that OCT-A data have a limited role in differentiating these disorders in the chronic phase.
Natural history of LHON has been already assessed using choroidal tickness in the study of Borelliand co-workers [34]. They found that macular and peripapillary choroidal thicknesses were significantly increased in the acute LHON stage. On the contrary, macular choroidal thickness was significantly reduced in the chronic stage.
The following references have been added:
- Kupersmith, M. J., Garvin, M. K., Wang, J. , Durbin, M., & Kardon, R. Retinal Ganglion Cell Layer Thinning Within One Month of Presentation for Non-Arteritic Anterior Ischemic Optic Neuropathy. Investigative ophthalmology & visual science. 2016, 57(8), 3588–3593.
- Kupersmith, M. J., Garvin, M. K., Wang, J. K., Durbin, M., & Kardon, R. Retinal ganglion cell layer thinning within one month of presentation for optic neuritis. Multiple sclerosis (Houndmills, Basingstoke, England). 2016, 22(5), 641–648.
- Pierro, L., Arrigo, A., Aragona, E., Cavalleri, M., & Bandello, F. Vessel Density and Vessel Tortuosity Quantitative Analysis of Arteritic and Non-arteritic Anterior Ischemic Optic Neuropathies: An Optical Coherence Tomography Angiography Study. Journal of clinical medicine. 2020, 9(4), 1094.
- Fard, M. A., Yadegari, S., Ghahvechian, H., Moghimi, S., Soltani-Moghaddam, R., & Subramanian, P. S. Optical Coherence Tomography Angiography of a Pale Optic Disc in Demyelinating Optic Neuritis and Ischemic Optic Neuropathy. Journal of neuro-ophthalmology : the official journal of the North American Neuro-Ophthalmology Society. 2019, 39(3), 339–344.
- Borrelli, E., Triolo, G., Cascavilla, M. L., La Morgia, C., Rizzo, G., Savini, G., Balducci, N., Nucci, P., Giglio, R., Darvizeh, F., Parisi, V., Bandello, F., Sadun, A. A., Carelli, V., & Barboni, P. Changes in Choroidal Thickness follow the RNFL Changes in Leber's Hereditary Optic Neuropathy. Scientific reports. 2016, 6, 37332.
- Conclusion
(Line 258-259) “Our study has shown that there is more decreased vasculature of the macula in LHON eyes than in NA-AION eyes in a chronic phase of the disease. “
Again, it is not indicated whether there was a statistical difference in these parameters between LHON and NA-AION. The authors noted that the lowest values of the superficial vessel density were obtained in LHON eyes and the lowest values of the deep vessel density were obtained in AION and AION fellow eyes. The meaning of the sentence in Line 258-259 is not clear.
In the abstract (line 29-30) it is now written as conclusion:
Perfusion and structure of the macula are affected both in chronic LHON and in chronic NA-AION with almost similar pattern.
Instead of:
Perfusion and structure of the macula, is affected more in chronic LHON than in chronic NA-AION.
In conclusion (line 395-396) it is now written:
Our study has shown that vasculature of the macula is decreased both in LHON and in NA-AION eyes in a chronic phase of the disease.
Instead of:
Our study has shown that there is more decreased vasculature of the macula in LHON eyes than in NA-AION eyes in a chronic phase of the disease.
- References
No. 6 and 23 seem to be the same paper.
Reference 23 has been deleted and the numbers of references have been changed.
Reviewer 2 Report
The paper is well prepared and written. The authors want to assess vasculature of the macula using OCT-A in patients with previous episode of Leber hereditary optic neuropathy (LHON) and non-arteritic anterior ischemic optic neuropathy (NA-AION). They found that perfusion and structure of the macula, is affected more in chronic LHON than in chronic NA-AION. Although the paper is well prepared and the study adds additional knowledge to the pathogenesis and natural history of LHON and NA-AION, one of the major limitations of the study is that the groups are relatively small. In addition, this is a cross-sectional study, longitudinal follow-ups are further required to monitor the changes of the the vasculature of optic neuropathies over time.
Author Response
Reviewer 2
The paper is well prepared and written. The authors want to assess vasculature of the macula using OCT-A in patients with previous episode of Leber hereditary optic neuropathy (LHON) and non-arteritic anterior ischemic optic neuropathy (NA-AION). They found that perfusion and structure of the macula, is affected more in chronic LHON than in chronic NA-AION. Although the paper is well prepared and the study adds additional knowledge to the pathogenesis and natural history of LHON and NA-AION, one of the major limitations of the study is that the groups are relatively small. In addition, this is a cross-sectional study, longitudinal follow-ups are further required to monitor the changes of the vasculature of optic neuropathies over time.
We would like to thank for this valuable comment. The limitations of the study are mentioned in the discussion chapter (line 390-393).
The shortcoming of our study is that the groups are relatively small, but in rare diseases as LHON is considered, only multicenter studies give possibility of larger groups of patients. As this is cross-sectional study, longitudinal studies are further required to monitor the changes of the the vasculature of optic neuropathies over time.

Round 2
Reviewer 1 Report
The authors investigated whether there is a difference in macular vasculature and retinal thickness in LHON and NAION in the chronic phase and whether OCTA can differentiate between the two. In the revised version, a direct comparison of LHON and NAION revealed statistically significant differences between the two in several parameters. However, the authors concluded that macular blood flow and structural damage were similar in LHON and NAION without fully discussing these differences. If the blood flow and structural damages in both diseases are similar, then one would conclude that OCTA has a limited role in differentiating these two diseases in the chronic phase. Yet, they stated that OCT-A may be additional valuable diagnostic tool to detect microvascular changes of the retina. In this paper, there is insufficient discussion of the results, and I feel uncomfortable with the conclusions.
A more detailed analysis of how macular blood flow and structure are impaired in both diseases and a deeper discussion of the results would make this study even more impressive. Hopefully this review will help to improve this study.
2.4. Statistical analysis
When both eyes of the same patient are included in the analysis, it is inherently necessary to account for intra-patient bias. For the control group, intra-patient bias was taken into account by using the mean of both eyes. However, the bias was not taken into consideration in the patient groups. Because LHON is a rare disease, the small sample size of this study is understandable. As I wrote in the previous review, I think it would be better to analyze only one eye per subject in both the patient and control groups, or use statistical methods that account for bias in both eyes of a same subject. However, if this is difficult, it should be stated in the limitation.
3.5. Comparison of superior and inferior hemifields of the macula
Was there a statistical difference in deep VD between superior and inferior hemifields in AION fellow eyes? (Table 5)
Since the sample size is small and authors are analyzing eyes with upper and lower damages together, it would be difficult to see differences between the hemifields in NAION group. It is not surprising that there is no difference in the LHON or control group between the superior and inferior hemifields. If you calculate the absolute value of the difference between the upper and lower hemifields of each patient and compare it between the groups, the difference may be greater in the NAION group.
3.6. Visual function results
3.7. Examples of cases
Line 217: Were there any NAION cases of visual field defects extending across the upper and lower visual fields? How was the VF of the new example NAION case in Figure 5?
>As statistical analysis revealed no significant differences between hemifields, in figure 5 a picture of the macula of patient with NA-AION has been replaced by more representative with no differences in the superior and inferior hemifield of the macula.
There is no need to change the representative cases.
According to the text, there was VF defects in the upper hemifield in 4 eyes, in the lower hemifield in 5 eyes and in the lower quadrant in 3 eyes. If these subjects are analyzed together, it is not surprising that there is no difference between the upper and lower retina.
As in the original representative case, there will be differences in blood flow and retinal thickness between the affected and unaffected hemifields in NAION patients. In contrast, in LHON patients, the entire macular area would be uniformly affected. The novelty of this paper would be enhanced if you could show that in NAION patients, there is a localized reduction in blood flow and retinal thinning, whereas in LHON patients, there is a uniform reduction in blood flow and thinning throughout the entire macula. If so, it could be said that OCTA and OCT may be useful in differentiating NAION and LHON in the chronic phase.
4. Discussion
The authors have not given enough thought to the results of this study.
There was no difference in superficial vessel density between LHON and NAION, whereas inner retinal layer thickness was thinner in LHON in some areas. In addition, deep VD was lower in NAION than in LHON. What are your thoughts on these results? The authors concluded in the Abstract that perfusion and structure of the macula are affected both in chronic LHON and in NA-AION with almost similar pattern. Is it so?
5. Conclusions
Line 343~:The authors stated that OCT-A may be additional valuable diagnostic tool both in LHON and NA-AION. However, how can OCTA be a valuable diagnostic tool if, as the authors conclude, the macular vasculature is equally affected in both diseases? It may be difficult to draw this conclusion from the current results.
Reviewer 2 Report
The paper is well prepared. The authors have answered the difficultuies they met during analysis the cases.